# LaTtE-Flow: Layerwise Timestep-Expert Flow-based Transformer

## Abstract

Recent advances in multimodal foundation models unifying image understanding and generation have opened exciting avenues for tackling a wide range of vision-language tasks within a single framework. Despite progress, existing unified models typically require extensive pretraining, and many of these models suffer from slow image generation speeds, limiting their practical deployment in real-time or resource-constrained settings. In this work, we propose **Layerwise Timestep-Expert Flow-based Transformer** (**LaTtE-Flow**), a novel architecture that improves the efficiency of diffusion/flow-based transformer within the unified model setting. LaTtE-Flow builds upon powerful pretrained Vision-Language Models (VLMs) to inherit strong multimodal understanding capabilities, and extends them with a novel Layerwise Timestep Experts flow-based architecture for efficient image generation. LaTtE-Flow distributes the flow-matching process across specialized groups of Transformer layers, each responsible for a distinct subset of timesteps. This design significantly improves sampling efficiency by activating only a small subset of layers at each sampling timestep. To further enhance performance, we propose a Timestep-Conditioned Residual Attention mechanism for efficient information reuse across layers. Experiments demonstrate that LaTtE-Flow achieves strong performance on multimodal understanding tasks, while achieving competitive image generation quality with around **6×** faster inference speed compared to recent unified multimodal models.

## 1 Introduction

Recent advances in multimodal foundation models capable of both image understanding and generation have opened promising avenues for building unified architectures that support a wide range of vision-language tasks (Shi et al., 2024; Wang et al., 2024b; Xie et al., 2025; Zhou et al., 2025; Chen et al., 2025c; Ma et al., 2025; Tong et al., 2024). Such unified multimodal models hold great potential for building general-purpose agents that can interpret, reason about, and generate multimodal content in response to user instructions. Current approaches to unified multimodal modeling generally fall into two broad categories. The first category leverages vector-quantized autoencoders (Van Den Oord et al., 2017; Esser et al., 2021; Yu et al., 2022) to discretize images into token sequences, which are then incorporated into the vocabulary of Large Language Models (LLMs) (Sun et al., 2024; Wang et al., 2024b; Xie et al., 2025; Wu et al., 2025a; Chen et al., 2025c; Wu et al., 2025b). These models are subsequently trained to autoregressively generate the next token, either textual or visual, thus integrating vision and language generation within a single framework. The second category leverages diffusion-based methods, either by coupling LLMs with external diffusion modules or by training LLMs to directly perform denoising steps (Zhou et al., 2025; Shi et al., 2024; Ma et al., 2025; Tong et al., 2024; Ge et al., 2024).

Despite significant progress, existing unified multimodal models still require extensive pretraining and struggle to support both multimodal understanding and image generation in an effective and efficient manner within a single architecture (Shen et al., 2025; Xiong et al., 2025). For example, unified models that leverage diffusion or flow-matching processes require dozens of forward passes through the full backbone during inference, resulting in slow and resource-intensive generation (Shen et al., 2025). Similarly, autoregressive approaches suffer from long decoding times, especially for high-resolution images that require generating large numbers of tokens sequentially (Xiong et al., 2025).

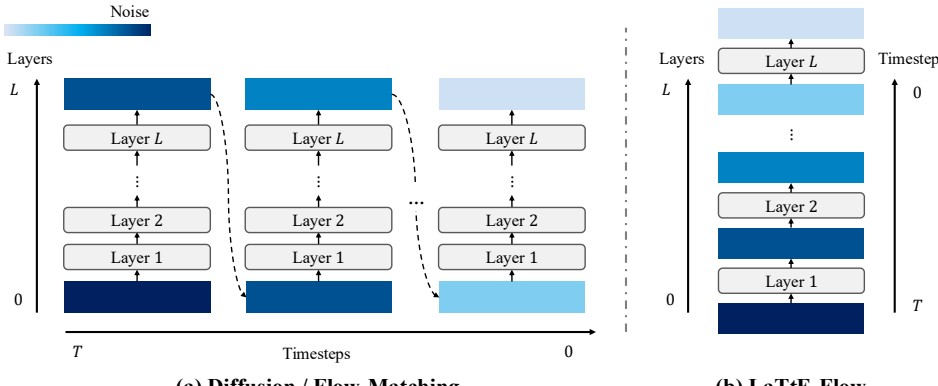

**(a) Diffusion / Flow-Matching**                    **(b) LaTtE-Flow**

Figure 1: **Flow-matching process between standard diffusion / flow-matching vs. our proposed LaTtE-Flow.** Unlike diffusion / flow-matching based models, which invoke the entire model at each sampling timestep, LaTtE-Flow activates only a subset of layers at each step, improving efficiency.

To address these challenges, we propose **Layerwise Timestep-Expert Flow-based Transformer (LaTtE-Flow)**, a novel architecture that improves the efficiency of diffusion/flow-based transformer within the unified model setting. In particular, LaTtE-Flow builds upon existing pre-trained VLMs that already possess strong multimodal understanding capabilities, and further introduces two key architectural innovations designed to enable efficient and high-quality image generation. First, we propose a novel **Layerwise Timestep Expert architecture**, which reduces the sampling time complexity by distributing the flow-matching process across groups of transformer layers. Instead of invoking the entire model across all time steps, LaTtE-Flow partitions transformer layers into disjoint groups, each assigned to a specific range of timesteps in the flow-matching process, as shown in Figure 1. During inference, only the relevant expert group is activated at each timestep, which drastically reduces computation while preserving generation quality. Second, we introduce **Timestep-Conditioned Residual Attention**, a lightweight mechanism that enables later layers to reuse self-attention maps computed at earlier layers, modulated by the current timestep. This design encourages the model to gradually refine features across layers, resulting in faster convergence during training. Experiments demonstrate that these two innovations enable LaTtE-Flow to achieve efficient and high-quality image generation. For example, LaTtE-Flow attains competitive generation quality with around 6× faster inference compared to recent unified models on ImageNet Deng et al. (2009), while maintaining strong multimodal understanding performance across several benchmark datasets. Extensive ablation studies highlight that LaTtE-Flow accelerates convergence and inference while preserving strong generation quality.

In summary, our contributions are: **(1)** We propose LaTtE-Flow, an efficient and unified multimodal architecture that integrates flow-matching-based image generation with pre-trained vision-language models. **(2)** We introduce a Layerwise Timestep Expert, a novel design that significantly reduces inference complexity by distributing transformer layers into timestep-specific experts. **(3)** We design a Timestep-Conditioned Residual Attention module, which enables effective reuse of attention information across layers, boosting training efficiency and performance. **(4)** Extensive experiments demonstrate that LaTtE-Flow achieves competitive performance on both generation and understanding tasks, while offering 6× faster inference compared to recent unified models.

## 2  RELATED WORK

**Unified Models.**    Unified multimodal architectures integrate multimodal understanding and generation within a single model, enabling general-purpose agents that can interpret and generate multimodal content in response to user instructions (Shi et al., 2024; Wang et al., 2024b; Xie et al., 2025; Zhou et al., 2025; Chen et al., 2025c; Ma et al., 2025; Tong et al., 2024). Existing approaches to unified modeling primarily fall into two categories: The first class of models relies on vector-quantized autoencoders Van Den Oord et al. (2017); Esser et al. (2021); Yu et al. (2022) to convert images into discrete token sequences that can be processed similarly to text. These visual tokens are added to the LLM vocabulary to enable unified autoregressive training over both language and vision (Sun et al., 2024; Wang et al., 2024b; Xie et al., 2025; Wu et al., 2025a; Chen et al., 2025c; Wu et al., 2025b).

The second class incorporates continuous generative processes, most notably diffusion models (Ho et al., 2020) or flow-matching models (Lipman et al., 2023). Some approaches connect LLMs with external diffusion modules, using the language model to guide image generation (Tong et al., 2024; Ge et al., 2024; Pan et al., 2025; Chen et al., 2025a; Xu et al., 2025), while others directly train LLMs to jointly perform denoising or flow-matching steps (Zhou et al., 2025; Shi et al., 2024; Ma et al., 2025). Despite progress in both categories, many of these models suffer from slow image generation speeds, limiting their practical deployment in real-time or resource-constrained settings.

**Multiple Experts in Diffusion Models.** Recent advancements in diffusion models have increasingly adopted modular or expert-based architectures for better image generation Sun et al. (2025); Shi et al. (2025). Building on this direction, several recent approaches have explored the use of expert models tailored to different diffusion timesteps (Lee et al., 2024; Fang et al., 2024; Zhuang et al., 2025). By allocating distinct experts to specific temporal intervals, these models aim to better capture the evolving nature of the denoising process. This design is partly motivated by findings from prior work Hang et al. (2023); Balaji et al. (2022), which show that optimization gradients from different timesteps often conflict, leading to slower convergence and degraded model performance. However, these models typically maintain a near full-parameter expert network for different timestep intervals, which leads to little or no improvement in inference efficiency under a fixed number of sampling steps. In contrast, we introduce a layerwise timestep expert architecture, which partitions the transformer layers into different groups of layers, each responsible for a specific range of timesteps. At inference time, only the corresponding group is activated, significantly reducing the number of parameters involved at each step. Moreover, our design allows all expert groups to be trained jointly, and we further integrate it within a unified model architecture, enhancing both efficiency and performance.

## 3 PRELIMINARIES

**Flow-Matching.** Flow-based generative models (Lipman et al., 2023; Liu et al., 2023; Albergo & Vanden-Eijnden, 2023) aim to learn a time-dependent velocity field $v_t$ that transports samples from a simple source distribution $p_0(x)$ (e.g., standard Gaussian) to a complex target distribution $p_1(x)$ via an ordinary differential equation (ODE):

$$\frac{dx_t}{dt} = v_t(x_t), \quad x_0 \sim p_0(x). \tag{1}$$

Recently, Lipman et al. (2023) propose a simple simulation-free Conditional Flow Matching (CFM) objective by defining a conditional probability path $p_t(x_t \mid x_1)$ and the corresponding conditional vector field $u_t(x_t \mid x_1)$ per sample $x_1$. The model directly regresses the velocity $v_t$ on a conditional vector field $u_t(\cdot \mid x_1)$:

$$\mathbb{E}_{t,p_1(x_1),p_t(x_t|x_1)}\|v_t(x_t,t) - u_t(x_t \mid x_1)\|^2, \tag{2}$$

where $u_t(\cdot \mid x_1)$ uniquely determines a conditional probability path $p_t(\cdot \mid x_1)$ towards target data sample $x_1$. A widely adopted choice for the conditional probability path is linear interpolation between the source and target data (Liu et al., 2023): $x_t = tx_1 + (1-t)x_0$. Assuming the source distribution $p_0$ is a standard Gaussian, this yields $x_t \sim \mathcal{N}(tx_1, (1-t)^2 I)$. Sampling from the learned model is obtained by sampling $x_0 \sim \mathcal{N}(x \mid 0, 1)$ and then numerically solving the ODE in Eq. (1).

## 4 LATTE-FLOW

We present LaTtE-Flow (Layerwise Timestep-Expert Flow-based Transformer), a novel architecture designed for efficient and high-quality image generation and multimodal understanding, unified within a single model. Built on top of pretrained Vision-Language Models (VLMs), LaTtE-Flow leverages their powerful understanding capabilities while introducing additional flow-matching based generation components to enable scalable and effective image synthesis. As illustrated in Figure 2, LaTtE-Flow is implemented as a mixture-of-transformer architecture, allowing for effective interaction between image latents and multimodal context. We also explore alternative architecture variants using a single transformer as the backbone in Appendix A, to highlight that our proposed method is not restricted to a single form.

Furthermore, we introduce two core architectural innovations applicable to both variants to enhance image generation efficiency and quality: **(1) Layerwise Timestep Experts** (Section 4.2), which partition the model into timestep-specialized modules to reduce sampling complexity, and **(2) Timestep-Conditioned Residual Attention** (Section 4.3), which injects timestep-aware residual attention into each attention layer through gating mechanisms modulated by a learned timestep embedding, improving training efficiency through effective information reuse across layers.

## 4.1 LaTtE-Flow Layer Design

LaTtE-Flow preserves the pretrained VLM entirely, keeping its parameters frozen (shown in **purple** in Figure 2) to retain strong multimodal understanding without finetuning. To enable image generation, it introduces a trainable generative pathway alongside the frozen backbone. Specifically, each Transformer layer is augmented with a trainable replica of the original VLM layer, along with additional components for flow-matching-based generation (shown in **blue** in Figure 2). LaTtE-Flow thus allows the model to perform image synthesis while leveraging the robust understanding capabilities of the pretrained VLM.

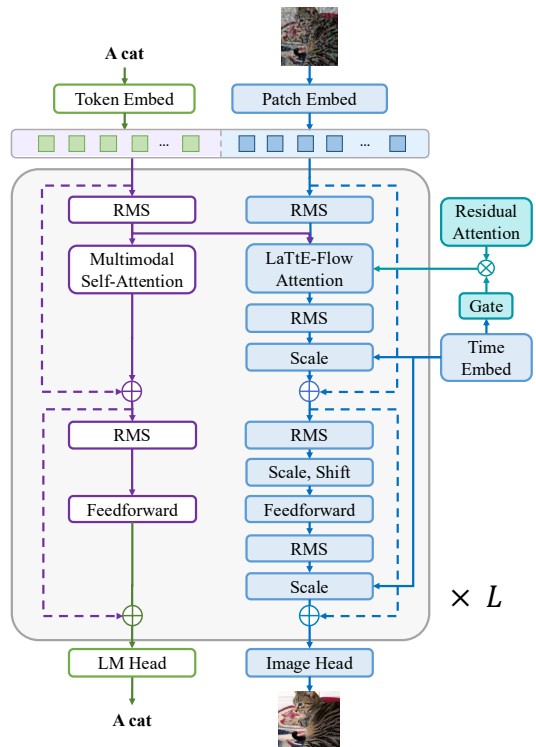

As illustrated in Figure 2, we introduce a LaTtE-Flow Attention module to enable effective interaction between generative image latents and multimodal context. Specifically, the noisy image latents—used during the flow-based generation process—attend to the text and visual context tokens, as detailed in Appendix B. This attention module employs a hybrid positional encoding scheme, combining the original 3D Rotary Positional Embeddings (RoPE) (Su et al., 2024), inherited from the pretrained VLM, for encoding spatial and temporal structure in the multimodal context, with newly introduced 2D positional encodings applied to the generative image tokens.

Figure 2: **LaTtE-Flow** **overall architecture**.

## 4.2 Layerwise Timestep Experts

Typical sampling procedures in diffusion models (Song & Ermon, 2019; Ho et al., 2020) or flow-matching models (Lipman et al., 2023; Liu et al., 2023; Albergo & Vanden-Eijnden, 2023) require repeatedly invoking the full network across a large number of timesteps, leading to slow inference-time speed. For instance, consider a standard diffusion transformer (DiT) model (Peebles & Xie, 2023) with $L$ transformer layers. The effective computational cost for $T$ sampling steps is $\mathcal{O}(L \times T)$, as shown in Figure 1 (a). To alleviate this inefficiency, we introduce a novel Layerwise Timestep Expert architecture, which reduces the effective sampling time complexity by distributing the flow-matching process across groups of transformer layers.

Specifically, instead of executing the entire model at every timestep, we partition the $L$ transformer layers into $K$ non-overlapping groups, where each group specializes in denoising samples within a specific timestep interval, as illustrated in Figure 1 (b). This design effectively enables efficient sampling, as only a subset of the network needs to be executed at each timestep.

Let each expert group be denoted as $\mathcal{G}_k^{l,l+M} = \{l, l+1, \ldots, l+M\}$, consisting of $M = L/K$ consecutive layers (from layer $l$ to layer $l + M$). During training, each layer group learns to predict the velocity field over its assigned timestep interval $[t_k, t_{k+1}]$ using a layerwise flow-matching loss. Specifically, each layer group $\mathcal{G}_k^{l,l+M}$ receives the noisy latent image $x_t \in \mathbb{R}^{N_x \times d}$ along with the multimodal context $m^l$, derived from the preceding layer $l-1$, and predicts the velocity field $s_\theta(x_t, m^l, t)$.

Formally, for timestep $t \in [t_k, t_{k+1}]$, the layerwise flow-matching loss is defined as:

$$\mathcal{L}_t = \mathbb{E}_{t, p_1(\boldsymbol{x}_1), p_t(\boldsymbol{x}_t | \boldsymbol{x}_1)} \left\| \mathcal{G}_k^{l, l+M}(\boldsymbol{x}_t, \boldsymbol{m}^l, t) - \boldsymbol{u}_t(\boldsymbol{x}_t \mid \boldsymbol{x}_1) \right\|^2, \quad \text{for } t \in [t_k, t_{k+1}], \qquad (3)$$

where $\mathcal{G}_k^{l, l+M}(\cdot)$ denotes the prediction produced by the expert group and $\boldsymbol{u}_t(\boldsymbol{x}_t \mid \boldsymbol{x}_1)$ is the ground-truth velocity at timestep $t$. By training each group exclusively on its respective timestep interval, LaTtE-Flow encourages timestep specialization, allowing the model to learn timestep-specific representations across the flow-matching process.

**Inference.** Let $C_{\text{layer}}$ denote the average forward compute cost of one Transformer layer per step. At inference time with $T'$ sampling steps, for each timestep $t \in [t_k, t_{k+1}]$, LaTtE-Flow activates only the associated expert layer group $\mathcal{G}_k^{l, l+M}$ to perform a forward pass from layer $l$ to layer $l + M$. This process is repeated across all $T'$ timesteps, with only $M = L/K$ layers evaluated per step. The multimodal hidden states, required for conditioning at each transformer layer, are computed once at the start of the inference and cached for reuse across all timesteps. Given one-time caching cost $C_{\text{cache}}$, the total inference cost for LaTtE-Flow is $C_{\text{cache}} + T' \times M \times C_{\text{layer}}$. In contrast, conventional diffusion models or flow-matching models execute all $L$ layers at every step, with total inference cost $C_{\text{cache}} + T' \times L \times C_{\text{layer}}$. The resulting relative speedup $S$ is

$$S = \frac{\mathcal{C}_{\text{baseline}}}{\mathcal{C}_{\text{LaTtE-Flow}}} = \frac{C_{\text{cache}} + T' \times L \times C_{\text{layer}}}{C_{\text{cache}} + T' \times (L/K) \times C_{\text{layer}}} = \frac{K + \theta}{1 + \theta}, \quad \text{where } \theta = \frac{C_{\text{cache}}}{T' \times M \times C_{\text{layer}}}. \qquad (4)$$

Since the one-time cache cost $C_{\text{cache}}$ is typically negligible compared to the cumulative compute across all sampling iterations $T'$. As the number of sampling steps $T'$ grows, the one-time cache cost is amortized, i.e., $\theta \to 0$ and hence $S \to K$. The resulting speed up shows that LaTtE-Flow guarantees an asymptotic $K$-fold reduction in per-step compute cost, and a complexity reduction from $\mathcal{O}(L \times T')$ to $\mathcal{O}(M \times T')$.

### 4.3 TIMESTEP-CONDITIONED RESIDUAL ATTENTION

To facilitate information reuse across transformer layers and improve both training efficiency and generative performance, we propose Timestep-Conditioned Residual Attention, a novel mechanism that introduces adaptive residual connections between successive image attention layers based on the current timestep. Inspired by the success of residual connection in ResNet (He et al., 2016), this design allows later layers to reuse and refine the attention patterns computed in earlier layers, while dynamically controlling the influence of past attention through the current flow-matching timestep.

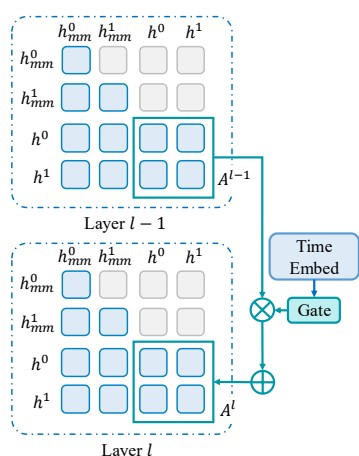

Let $\boldsymbol{A}^l \in \mathbb{R}^{N_x \times N_x}$ image self-attention matrix at layer $l$, where $N_x$ is the number of image tokens. In a standard self-attention layer, the attention matrix is computed as:

$$\boldsymbol{A} = \text{Softmax}\left( \frac{(\boldsymbol{h} \boldsymbol{W}^Q)(\boldsymbol{h} \boldsymbol{W}^K)^T}{\sqrt{d}} \right), \qquad (5)$$

Figure 3: **Timestep-conditioned residual attention**

where $\boldsymbol{h} \in \mathbb{R}^{N_x \times d}$ denotes the hidden states of the noisy image latents, and $\boldsymbol{W}^Q, \boldsymbol{W}^K \in \mathbb{R}^{d \times d}$ are learnable query and key projection matrices.

To incorporate residual attention from the previous layer, we define the augmented self-attention matrix at layer $l + 1$ as:

$$\tilde{\boldsymbol{A}}^{l+1} = \boldsymbol{A}^{l+1} + g(t) \odot \boldsymbol{A}^l, \quad g(t) = \tanh(\boldsymbol{h}_t \boldsymbol{W}_t), \qquad (6)$$

where $\boldsymbol{h}_t \in \mathbb{R}^d$ is the embedding of the current flow-matching timestep $t$ and $\boldsymbol{W}_t \in \mathbb{R}^{d \times H}$ is a trainable projection matrix, with $d$ denoting the hidden dimension and $H$ the number of attention heads.

The head-wise gating vector $g(t) \in (-1, 1)^H$, produced by a $\tanh(\cdot)$ activation, dynamically controls the extent to which each attention head incorporates residual attention information from the previous layer. The operator $\odot$ denotes element-wise multiplication, broadcast across all attention heads. Notably, while the LaTtE-Flow Attention module jointly processes both noisy image states and multimodal hidden states, the residual attention mechanism is applied only to the self-attention map over the noisy image hidden states, as shown in Figure 3.

The timestep-conditioned residual attention mechanism enables the model to dynamically control how much residual attention from the previous layer is incorporated into the current layer, on a per-head basis and conditioned on the timestep. Empirically, this design accelerates convergence during training and enhances the quality of generated images.

## 5 EXPERIMENT SETUP

**Backbone Model and Image Encoder.** LaTtE-Flow is built upon Qwen2-VL-2B-Instruct (Wang et al., 2024a), a pretrained VLM composed of $L = 28$ transformer layers. We create a trainable copy of each Transformer layer from the original Qwen2-VL-2B-Instruct and integrate it with additional components tailored for flow-matching-based image generation. These duplicated components are initialized with the corresponding pretrained weights from the original VLM. For image encoding, we adopt the recently proposed Deep Compression Autoencoder (DC-AE) Chen et al. (2025b), which compresses raw image pixels into a compact latent space using a 32× down-sampling ratio.

**Timestep Distribution.** To enable Layerwise Timestep Experts, LaTtE-Flow partitions the model into $K = 4$ non-overlapping layer groups, each containing $M = 7$ consecutive layers for the final results. These groups are designed to operate over distinct intervals of the flow-matching timesteps. During training, we use $T = 1000$ flow-matching steps, which are initially divided uniformly into four intervals. To encourage robustness near interval boundaries and promote smooth transitions across groups, we introduce a 100-step overlap between adjacent timestep intervals during training. This overlap allows boundary timesteps to be seen by multiple layer groups, improving generalization. At inference time, we disable the overlaps to maintain strict partitioning of timestep intervals. Consequently, at each denoising step, only the corresponding expert layer group is activated, requiring just $M = 7$ layers per inference step. This contrasts favorably with standard diffusion or flow-matching models that activate all $L = 28$ layers at every step, significantly enhancing generation efficiency. Further details are provided in Appendix C.

**Baseline Architectures.** We construct the baseline model Vanilla, which matches the architectures of LaTtE-Flow, but excludes both the Layerwise Timestep Experts and Timestep-Conditioned Residual Attention mechanisms, allowing us to directly evaluate the effectiveness of these proposed mechanisms. The Vanilla baseline retains a parallel generative path alongside the original VLM modules. Conceptually, it resembles prior models such as LMFusion (Shi et al., 2024), which augment language models with a separate branch for handling image generation.

**Training and Evaluation Details.** All LaTtE-Flow variants are trained on 1.2M images from the ImageNet Deng et al. (2009) training split at a resolution of $256 \times 256$ with a global batch size of 2048 and a constant learning rate of $5e\text{-}4$ for 240K steps. Instead of using class IDs for the ImageNet experiments, we use the corresponding natural language captions for both training and evaluation. For both Vanilla and LaTtE-Flow, we only fine-tune parameters specialized for image generation while keeping parameters for image understanding frozen. For evaluation, we report FID, Inception Score, Precision, and Recall on ImageNet following previous convention Peebles & Xie (2023). Additional details in Appendix C.

## 6 RESULTS AND DISCUSSION

### 6.1 IMAGE GENERATION AND UNDERSTANDING RESULTS

We evaluate LaTtE-Flow on both image generation (Table 4) and multimodal understanding (Table 2) tasks. Table 4 reports quantitative comparison between LaTtE-Flow, recent unified models, and leading image generation models. We evaluate each model in terms of generation quality, activated parameters for each inference step, and inference efficiency. All inference times are measured on a single NVIDIA L40 GPU with batch size 50. LaTtE-Flow achieves better FID scores compared

Table 1: **Comparison of generative models** across FID, IS, Precision, Recall, parameters, steps, and inference time on ImageNet-50K. For LaTtE-Flow, we report the number of parameters activated per timestep, given that it has a timestep-expert architecture where only a subset of layers is used at each step. Rel. Time: inference time relative to LaTtE-Flow. †: taken from MaskGIT (Chang et al., 2022)

| | Model | FID↓ | IS↑ | Pre↑ | Rec↑ | #Params | #Step | Time (s / img) | Rel. Time |
|---|---|---|---|---|---|---|---|---|---|
| **Diffusion Models** | ADM (Dhariwal & Nichol, 2021) | 10.94 | 101.0 | 0.69 | 0.63 | 554M | 250 | 9.677 | 168 |
| | CDM (Ho et al., 2022) | 4.88 | 158.7 | – | – | – | 8100 | – | |
| | LDM-4-G (Rombach et al., 2022) | 3.60 | 247.7 | – | – | 400M | 250 | – | |
| | DiT-L/2 (Peebles & Xie, 2023) | 5.02 | 167.2 | 0.75 | 0.57 | 458M | 250 | 1.786 | 31 |
| | DiT-XL/2 (Peebles & Xie, 2023) | 2.27 | 278.2 | 0.83 | 0.57 | 675M | 250 | 2.592 | 45 |
| **Masked Models** | MaskGIT (Chang et al., 2022) | 6.18 | 182.1 | 0.80 | 0.51 | 227M | 8 | 0.029 | 0.5 |
| | MAGE (Li et al., 2023a) | 6.93 | 195.8 | – | – | 230M | – | – | |
| **AR Models** | VQVAE-2† (Razavi et al., 2019) | 31.11 | ~45 | 0.36 | 0.57 | 13.5B | 5120 | – | |
| | VQGAN† (Esser et al., 2021) | 18.65 | 80.4 | 0.78 | 0.26 | 227M | 256 | 1.094 | 19 |
| | VQGAN (Esser et al., 2021) | 15.78 | 74.3 | – | – | 1.4B | 256 | 1.382 | 24 |
| | ViT-VQGAN (Yu et al., 2022) | 4.17 | 175.1 | – | – | 1.7B | 1024 | 1.382 | 24 |
| | RQTran. (Lee et al., 2022) | 7.55 | 134.0 | – | – | 3.8B | 68 | 1.210 | 21 |
| **Unified Models** | Show-o (Xie et al., 2025) | 31.26 | 98.7 | 0.55 | 0.69 | 1.3B | 50 | 2.493 | 48 |
| | Janus Pro (Chen et al., 2025c) | 23.68 | 105.2 | 0.58 | 0.49 | 1.5B | 576 | 0.311 | 6 |
| | Vanilla (Ours) | 6.33 | 192.4 | 0.80 | 0.67 | 2.0B | 40 | 0.158 | 3 |
| | LaTtE-Flow (Ours) | 5.79 | 213.1 | 0.78 | 0.69 | 500M | 40 | 0.052 | 1 |

Table 2: **Results on comprehensive image understanding benchmarks.** Best scores are highlighted in **bold**. Since our LaTtE-Flow is an expert architecture, we report the number of activated parameters used for image understanding. LaTtE-Flow preserves Qwen2-VL-2B's strong understanding performance.

| Model | MMBench | SEED | POPE | MM-Vet | MME-P | MMMU | RWQA | TEXTVQA | #Params | TFLOPs |
|---|---|---|---|---|---|---|---|---|---|---|
| EMU2 Chat (Sun et al., 2024) | - | 62.8 | - | 48.5 | - | 34.1 | - | 66.6 | 34B | 5.4 |
| Chameleon (Team, 2024) | 19.8 | 27.2 | 19.4 | 8.3 | 202.7 | 22.4 | 39.0 | 0.0 | 7B | 3.6 |
| Chameleon (Team, 2024) | 32.7 | - | 59.8 | 9.7 | 604.5 | 38.8 | 39.2 | 0.0 | 34B | 17.4 |
| Seed-X (Ge et al., 2024) | 70.1 | 66.5 | 84.2 | 43.0 | 1457.0 | 35.6 | - | - | 17B | 11.1 |
| VILA-U (Wu et al., 2025b) | 66.6 | 57.1 | 85.8 | 33.5 | 1401.8 | 32.2 | 46.6 | 48.3 | 7B | 3.6 |
| EMU3 (Wang et al., 2024b) | 58.5 | 68.2 | 85.2 | 37.2 | 1243.8 | 31.6 | 57.4 | 64.7 | 8B | 4.1 |
| MetaMorph (Tong et al., 2024) | 75.2 | 71.8 | - | - | - | **41.8** | 58.3 | 60.5 | 8B | 1.1 |
| Show-o (Xie et al., 2025) | - | - | 80.0 | - | 1097.2 | 27.4 | - | - | 1.3B | 0.7 |
| Janus (Wu et al., 2025a) | 69.4 | 63.7 | 87.0 | 34.3 | 1338.0 | 30.5 | - | - | 1.5B | 0.8 |
| Janus Pro (Chen et al., 2025c) | **75.5** | 68.3 | 86.2 | 39.8 | 1444.0 | 36.3 | - | - | 1.5B | 0.8 |
| Qwen2-VL-2B (Wang et al., 2024a) | 74.9 | **72.4** | **87.3** | **51.5** | **1501.4** | 41.1 | **60.7** | **79.7** | 2B | 0.4 |
| LaTtE-Flow | 74.9 | **72.4** | **87.3** | **51.5** | **1501.4** | 41.1 | **60.7** | **79.7** | 2B | 0.4 |

to state-of-the-art unified models Xie et al. (2025); Wu et al. (2025a); Chen et al. (2025c) that are pretrained on the mixture of ImageNet and other large-scale image-caption datasets, while achieving much faster inference speed, i.e., 48× faster than Show-o Xie et al. (2025) and 6× faster than Janus Pro Chen et al. (2025c). Moreover, LaTtE-Flow outperforms its respective baselines, Vanilla, which are conceptually similar to LMFusion Shi et al. (2024), with much fewer activated parameters per flow-matching step and 3× faster inference speed. The computational cost of Vanilla is 28.3 TFLOPs per forward pass, compared to only 7.08 TFLOPs for LaTtE-Flow, further underscoring the efficiency of the proposed method. In addition, LaTtE-Flow exhibits competitive performance compared to diffusion models Dhariwal & Nichol (2021); Ho et al. (2022); Rombach et al. (2022); Peebles & Xie (2023), Masked Models Chang et al. (2022); Li et al. (2023a) and Auto-regressive (AR) models Razavi et al. (2019); Esser et al. (2021); Yu et al. (2022); Lee et al. (2022) that are specialized for image generation, achieving better parameter and inference-time efficiency. These results suggest LaTtE-Flow as a promising, efficient, and effective architecture for image generation. Qualitative results on ImageNet are provided in Appendix E.

Table 2 presents results on multimodal understanding benchmarks Liu et al. (2024); Li et al. (2024; 2023b); Yu et al. (2024); Fu et al. (2023); Yue et al. (2024); Singh et al. (2019). LaTtE-Flow achieves competitive or superior performance compared to recent unified models. By effectively leveraging a frozen vision-language backbone, the understanding capability of LaTtE-Flow is inherited from its pretrained backbone model Qwen2-VL-2B-Instruct (Wang et al., 2024a), and therefore matches the performance of the backbone itself. This approach aligns with concurrent

studies (Chen et al., 2025a; Lin et al., 2025), which also employ frozen backbones to fully exploit the pretrained understanding strength.

## 6.2 ABLATION STUDIES

**Faster Convergence Rate of LaTtE-Flow.** Figure 4 illustrates the training dynamics of LaTtE-Flow compared to Vanilla. We observe that LaTtE-Flow exhibits a significantly faster convergence rate during training, reaching competitive image generation performance (lower FID) in fewer training steps. We attribute this favorable property of LaTtE-Flow to the layerwise timestep-expert architecture. As noted in prior work Balaji et al. (2022); Hang et al. (2023), the slow convergence of diffusion models is partially due to the conflicting optimization directions of different timesteps. Optimizing for timesteps that are close can benefit each other, while optimizing timesteps that are far away can interfere with each other. LaTtE-Flow's

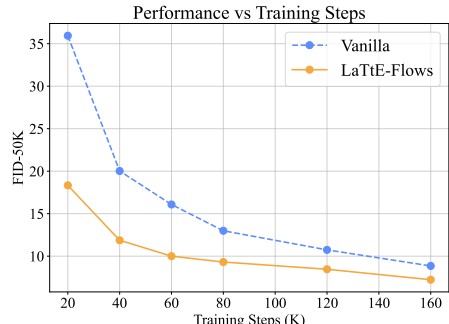

Figure 4: **Training dynamics of LaTtE-Flow vs. Vanilla.** FID on ImageNet 50K.

layerwise timestep-expert architecture alleviates this challenge by distributing timesteps across different transformer layers.

**Impact of Varying Group Size.** We also investigate how the timestep-expert group size $M$ affects the trade-off between generation quality and inference efficiency. Specifically, we train LaTtE-Flow with group sizes $M \in \{4, 7, 14\}$, corresponding to partitioning the transformer layers into 7, 4, and 2 expert groups, respectively. Figure 5 reports results at 120K training steps. We observe that larger group sizes consistently improve generation quality, as measured by FID, due to increased modeling capacity. However, this comes at the cost of reduced inference speed, since more layers are executed per timestep. Both $M = 7$ and $M = 14$ achieve better generation quality and efficiency compared to the baseline Vanilla (Vanilla), which applies all 28 layers at every step. Thus, considering the trade-off between performance and efficiency, we select $M = 7$ as the default group size in our main results in Table 4, which offers strong generation quality with substantial sampling speedups.

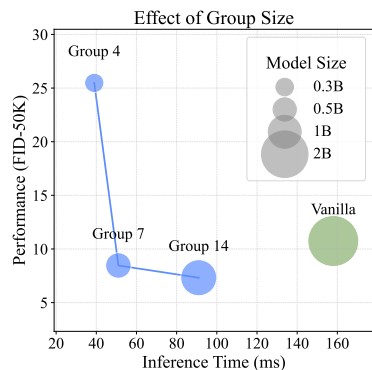

Figure 5: **Effect of group size in LaTtE-Flow**.

**Effect of Timestep-Conditioned Residual Attention.** To quantify the effect of timestep-conditioned residual attention, we compare LaTtE-Flow against a variant with the timestep-conditioned residual

Table 3: **Effect of time-conditioned residual attention.**

| Model | FID↓ | IS↑ | Pre↑ | Rec↑ |
|---|---|---|---|---|
| LaTtE-Flow | 5.79 | 213.1 | 0.78 | 0.69 |
| - w/o Residual Attention | 8.26 | 157.0 | 0.75 | 0.61 |

attention removed. As shown in Table 3, removing residual attention leads to a notable degradation across multiple metrics, highlighting the effectiveness of time-conditioned attention across layers. Adding timestep-conditioned residual attention does not introduce additional inference time cost.

**Effect of Sampling Steps and CFG.** Figure 6 shows the impact of varying the number of sampling steps and classifier-free guidance scale (CFG) on image generation quality. We observe that increasing the number of steps generally improves image generation quality, leading to lower FID and higher Inception Score. However, as the number of sampling steps surpasses 40, performance improvements become marginal. In general, higher CFG leads to better Inception Score, but for FID, once the CFG goes beyond 5, performance starts to decrease slightly.

**Timestep Condition in Residual Attention.** To better understand the role of timestep conditioning in residual attention, we perform an in-depth analysis on LaTtE-Flow. Specifically, we first investigate how attention patterns evolve across transformer layers and sampling timesteps in baseline models. We quantify the sequential similarity between adjacent layers at each timestep using a

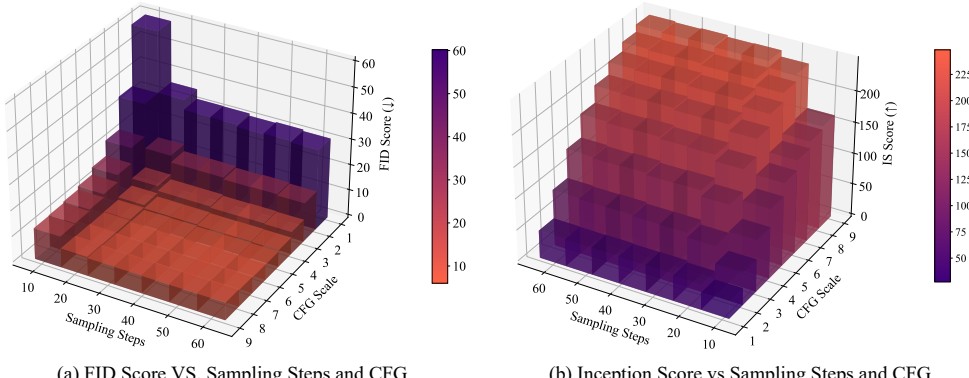

(a) FID Score VS. Sampling Steps and CFG      (b) Inception Score vs Sampling Steps and CFG

Figure 6: **Impact of # sampling steps and CFG strength on Inception Score and FID.**

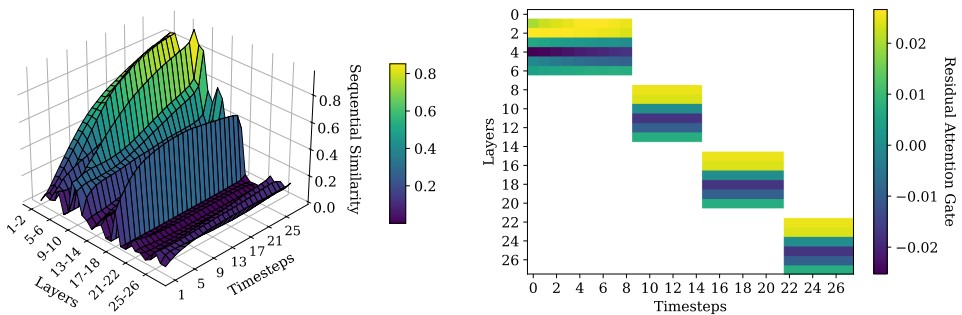

(a) Timestep-varying sequential similarity      (b) Timestep-conditioned residual attention
across adjacent transformer layers               gate for head 5 across transformer layers

Figure 7: **Timestep-conditioned residual attention analysis.** (a) Visualization of attention behavior in Vanilla and (b) learned residual gating patterns in LaTtE-Flow.

total variation-based metric:

$$S(\boldsymbol{A}^l, \boldsymbol{A}^{l+1}) = 1 - \frac{1}{2} \sum_i \left| \mathrm{Softmax}\left(\boldsymbol{A}_i^l\right) - \mathrm{Softmax}\left(\boldsymbol{A}_i^{l+1}\right) \right|, \tag{7}$$

where $\mathrm{Softmax}\left(\boldsymbol{A}_i^l\right)$ is the softmax-normalized $i$-th row of attention map $\boldsymbol{A}^l$. Higher values of $S$ reflect greater similarity in image attention maps between successive layers.

Figure 7 (a) shows how sequential similarity in Vanilla evolves throughout the sampling process, averaged over 100 randomly selected samples. We observe that early in sampling, attention maps across layers show low similarity, but as generation progresses, especially in later timesteps, similarity increases, sometimes approaching 1.0 in early layers. This motivates using residual attention for efficient reuse, with dynamic gating needed to adapt to varying similarity patterns across timesteps. Figure 7 (b) shows timestep-conditioned residual attention gates in LaTtE-Flow, which modulate how much past-layer attention is reused. As seen across all heads (Figure 14), gating remains stable across timesteps within a head but varies between heads, indicating specialization. These results highlight the effectiveness of dynamic, head-specific residual attention in flow-matching generation.

## 7 CONCLUSION

In this work, we present Layerwise Timestep-Expert Flow-based Transformer (LaTtE-Flow), a novel architecture that improves the efficiency of diffusion/flow-based transformer within the unified model setting. LaTtE-Flow introduces two key novel architectural innovations: **Layerwise Timestep Experts**, which reduces sampling complexity by specializing transformer layers to distinct timestep intervals, and **Timestep-Conditioned Residual Attention**, which facilitates adaptive reuse and refinement of attention structures across layers. Extensive experimental evaluations demonstrate that LaTtE-Flow not only achieves strong multimodal understanding and image generation performance, but also achieves around 6× faster inference compared to existing unified models.

## REPRODUCIBILITY STATEMENT

We will fully release the source code and the trained model weights to facilitate reproducibility. Detailed implementation settings for both training and evaluation are provided in Section 5, with additional specifications included in Appendix C.

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

## A  LaTtE-Flow Blend

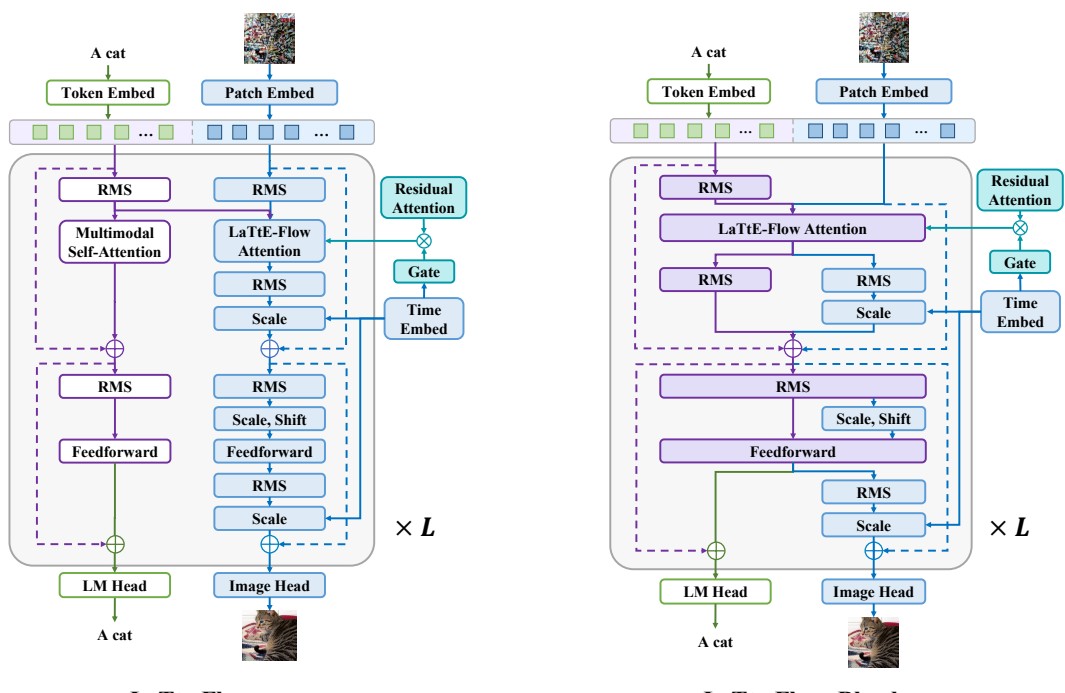

Figure 8: **LaTtE-Flow overall architecture**.

To demonstrate that LaTtE-Flow is not tied to a specific flow-matching architecture, we also introduce **LaTtE-Flow Blend** and apply our method on the Blend architecture as well. Figure 8 shows that LaTtE-Flow Blend unifies the image generation and understanding components through a partially shared transformer layer. Here, each layer consists of task-specific submodules with separate parameters for generation and understanding, and a set of shared submodules that are used by both tasks. This design enables tighter fusion between generation and understanding signals, facilitating more effective information exchange while maintaining flexibility to specialize for each modality.

We also construct the baseline model Vanilla Blend, which matches the architectures of LaTtE-Flow Blend, but excludes both the Layerwise Timestep Experts and Timestep-Conditioned Residual Attention mechanisms, allowing us to directly evaluate the effectiveness of these proposed mechanisms on different architecture. The Vanilla Blend baseline unified generation and understanding computations within shared layers, akin to the design of Transfusion (Zhou et al., 2025). And we perform a full parameter fine-tuning for Vanilla Blend and LaTtE-Flow Blend.

Table 4 reports quantitative comparison between Vanilla Blend, LaTtE-Flow Blend, recent unified models, and leading image generation models. We show that both LaTtE-Flow variants outperform their respective baselines, Vanilla Blend and Vanilla, which are conceptually similar to Transfusion Zhou et al. (2025) and LMFusion Shi et al. (2024), with much fewer activated parameters per flow-matching step and 3 to 4× faster inference speed.

## B  LaTtE-Flow Attention Module

Figure 9 illustrates the architecture of the LaTtE-Flow Attention module. Our framework applies 3D Rotary Positional Embeddings (RoPE) (Su et al., 2024) from the pretrained VLM to multimodal hidden states and uses a new 2D Rotary Positional Embeddings to the generative image tokens. We adopt bi-directional attention on generative image tokens, and all generative image tokens are allowed to attend to previous multimodal tokens.

Table 4: **Comparison of generative models** across FID, IS, Precision, Recall, parameters, steps, and inference time on ImageNet-50K. For LaTtE-Flow, we report the number of parameters activated per timestep, given that it has a timestep-expert architecture where only a subset of layers is used at each step. Rel. Time: inference time relative to LaTtE-Flow. †: taken from MaskGIT (Chang et al., 2022)

| | Model | FID↓ | IS↑ | Pre↑ | Rec↑ | #Params | #Step | Time (s / img) | Rel. Time |
|---|---|---|---|---|---|---|---|---|---|
| **Diffusion Models** | ADM (Dhariwal & Nichol, 2021) | 10.94 | 101.0 | 0.69 | 0.63 | 554M | 250 | 9.677 | 168 |
| | CDM (Ho et al., 2022) | 4.88 | 158.7 | – | – | – | 8100 | – | |
| | LDM-4-G (Rombach et al., 2022) | 3.60 | 247.7 | – | – | 400M | 250 | – | |
| | DiT-L/2 (Peebles & Xie, 2023) | 5.02 | 167.2 | 0.75 | 0.57 | 458M | 250 | 1.786 | 31 |
| | DiT-XL/2 (Peebles & Xie, 2023) | 2.27 | 278.2 | 0.83 | 0.57 | 675M | 250 | 2.592 | 45 |
| **Masked Models** | MaskGIT (Chang et al., 2022) | 6.18 | 182.1 | 0.80 | 0.51 | 227M | 8 | 0.029 | 0.5 |
| | MAGE (Li et al., 2023a) | 6.93 | 195.8 | – | – | 230M | – | – | |
| **AR Models** | VQVAE-2$^\dagger$ (Razavi et al., 2019) | 31.11 | ~45 | 0.36 | 0.57 | 13.5B | 5120 | – | |
| | VQGAN$^\dagger$ (Esser et al., 2021) | 18.65 | 80.4 | 0.78 | 0.26 | 227M | 256 | 1.094 | 19 |
| | VQGAN (Esser et al., 2021) | 15.78 | 74.3 | – | – | 1.4B | 256 | 1.382 | 24 |
| | ViT-VQGAN (Yu et al., 2022) | 4.17 | 175.1 | – | – | 1.7B | 1024 | 1.382 | 24 |
| | RQTran. (Lee et al., 2022) | 7.55 | 134.0 | – | – | 3.8B | 68 | 1.210 | 21 |
| **Unified Models** | Show-o (Xie et al., 2025) | 31.26 | 98.7 | 0.55 | 0.69 | 1.3B | 50 | 2.493 | 48 |
| | Janus Pro (Chen et al., 2025c) | 23.68 | 105.2 | 0.58 | 0.49 | 1.5B | 576 | 0.311 | 6 |
| | Vanilla Blend (Ours) | 6.12 | 193.7 | 0.78 | 0.69 | 2.0B | 40 | 0.185 | 4 |
| | LaTtE-Flow Blend (Ours) | 6.03 | 193.9 | 0.77 | 0.68 | 500M | 40 | 0.061 | 1 |
| | Vanilla (Ours) | 6.33 | 192.4 | 0.80 | 0.67 | 2.0B | 40 | 0.158 | 3 |
| | LaTtE-Flow (Ours) | 5.79 | 213.1 | 0.78 | 0.69 | 500M | 40 | 0.052 | 1 |

**LaTtE-Flow Attention**

Figure 9: **LaTtE-Flow Attention**

## C IMPLMENTATION DETAILS

**Timestep Distribution.** To enable Layerwise Timestep Experts, LaTtE-Flow partitions the model into $K = 4$ non-overlapping layer groups, each containing $M = 7$ consecutive layers for the final results. These groups are designed to operate over distinct intervals of the flow-matching timesteps. During training, we use $T = 1000$ flow-matching steps, which are initially divided uniformly into four intervals: $[1000.0, 750.25]$, $[750.25, 500.50]$, $[500.50, 250.75]$, and $[250.75, 0]$. To encourage robustness near interval boundaries and promote smooth transitions across groups, we introduce a 100-step overlap between adjacent timestep intervals during training. This overlap allows boundary timesteps to be seen by multiple layer groups, improving generalization. Specifically, layers 1 through 7 are assigned to the timestep interval $[1000, 700]$, layers 8 through 14 cover $[700, 450]$, layers 15 through 21 operate on $[450, 200]$, and layers 22 through 28 handle the final interval $[200, 0]$. Each group is trained exclusively on its assigned range according to Eq. (3), enabling it to specialize in the velocity prediction of that particular segment of the flow-matching timestep interval.

At inference time, we disable overlaps to maintain strict partitioning of timestep intervals. Consequently, at each denoising step, only the corresponding expert layer group is activated, requiring just

$M = 7$ layers per inference step. This contrasts favorably with standard diffusion or flow-matching models that activate all $L = 28$ layers at every step, significantly enhancing generation efficiency.

**Training and Evaluation Details.** We train all model variants on eight H200 for approximately four days. During training, following previous approaches, we employ classifier-free guidance (Ho & Salimans, 2022) to guide the sampling process for better sampling quality by amplifying the difference between conditional and unconditional generation with the guidance scale > 1. During training, we randomly drop the multimodal condition with probability $10\%$ to facilitate unconditional prediction.

For evaluation, each model generates 50 images for each of 1,000 classes in ImageNet with 40 sampling steps and classifier-free guidance (CFG) of 5 based on our ablation study in Section 6.2. We report FID and Inception Score of 50K generated images against 50K real images from the ImageNet validation split. Following previous convention Peebles & Xie (2023), we compute Precision and Recall using 1,000 generated images. All scores are calculated using standard implementations from torch-fidelity [1].

# D USER STUDY

To complement the automated metrics and further assess the generative quality of LaTtE-Flow, we conduct a human preference study comparing our model against two recent unified model baselines, Janus Pro (Chen et al., 2025c) and Show-o (Xie et al., 2025). We randomly sample 50 class prompts from ImageNet and generate images for each prompt using all three models. For each prompt, we present the three corresponding images to human evaluators in randomized order to avoid positional bias. We recruit 10 annotators and instruct them to select the image they prefer, with explicit guidance to evaluate along two axes: (1) *photo-realism*, and (2) *semantic accuracy* with respect to the prompt. The full annotation guideline is:

```
Please follow the instructions below when evaluating images:

Please do not rely solely on overall image aesthetic quality (e.g.,
style, beauty, artistic appeal) when determining preference. You should
also pay attention to photo-realism, as ImageNet-1k consists of photo-
realistic images.

In addition, a model may generate a visually impressive image that is
semantically incorrect. Please carefully verify that the main object or
animal in the image matches the caption. Check for correct species and
object identity as described on the left.

Your evaluation should be based primarily on:

1. Photo-realism
2. Semantic accuracy (whether the visual content truly corresponds to
the caption)

For each row in the table, you will see three images generated by
different models for the same caption.
Please rank the images (1 = best, 3 = worst).

You may assign ties if multiple images are equally good or equally bad.
For example: 1, 1, 2 → two best images tie for rank 1.
```

Figure 10, reports the win, tie, and loss rates of LaTtE-Flow compared to the baselines. LaTtE-Flow is preferred to Janus Pro in 71.4% of cases (with 8.6% ties and 19.6% losses) and preferred to Show-o in 63.4% of cases (with 5.0% ties and 31.4% losses).

---

[1] https://github.com/toshas/torch-fidelity

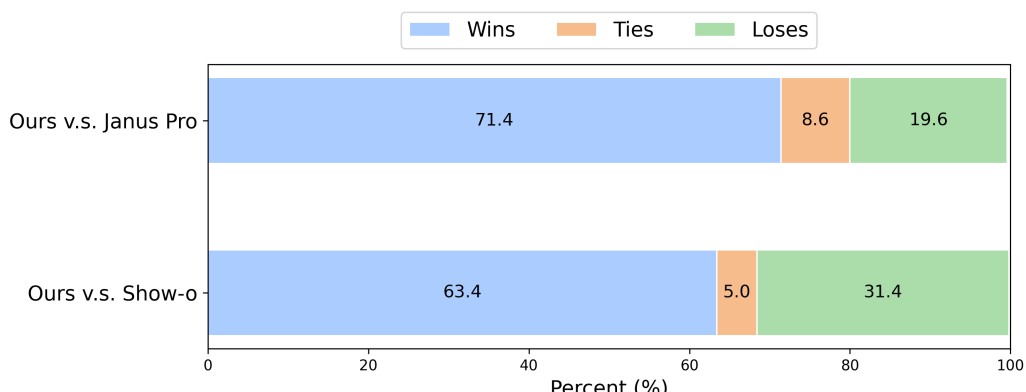

Figure 10: **Human preference study results.** We report pairwise win/tie/loss rates between LaTtE-Flow and each baseline.

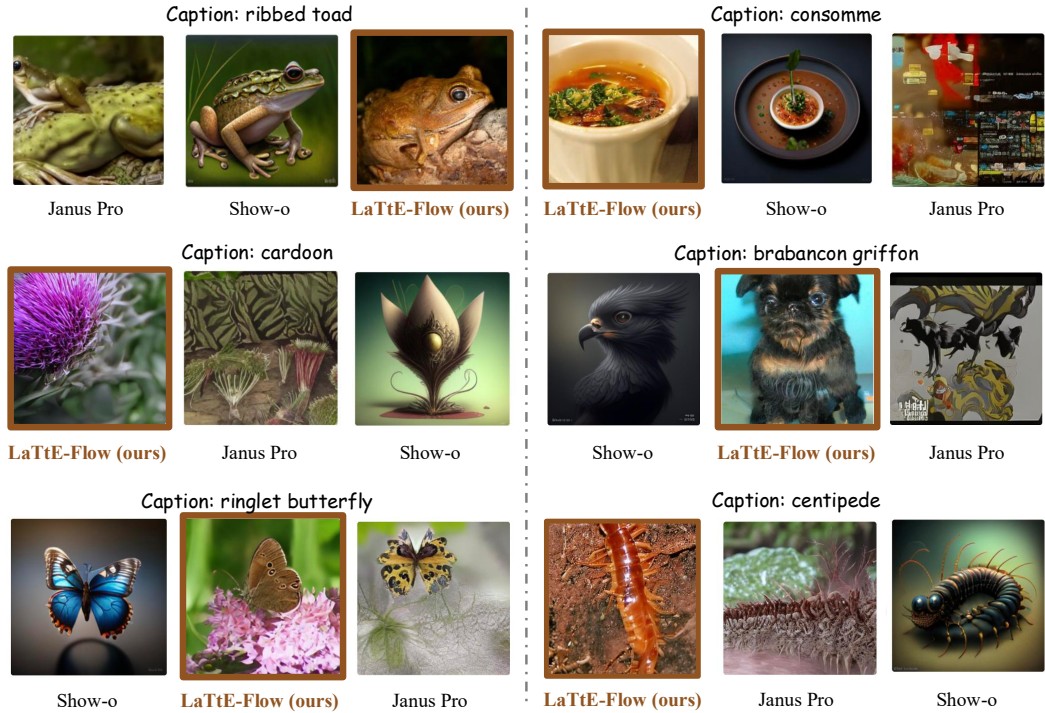

Figure 11: **Qualitative examples of user-study comparisons.** For visualization purposes, we display the model names below each image and highlight the output of LaTtE-Flow using a brown frame. Note that in the actual user study, all generated images were anonymized and unframed to avoid revealing model identity or introducing positional bias.

Moreover, Figure 11 presents several qualitative comparison examples used in the study. As shown, Show-o sometimes produces visually appealing images but fails to align with the given prompt. Janus Pro, on the other hand, tends to generate images in which the target object loses structural integrity. In contrast, LaTtE-Flow is able to produce images that are both photo-realistic and semantically faithful to the prompt.

## E    QUALITATIVE RESULTS

Figure 12 shows the qualitative results of sampled $256 \times 256$ images by LaTtE-Flow.

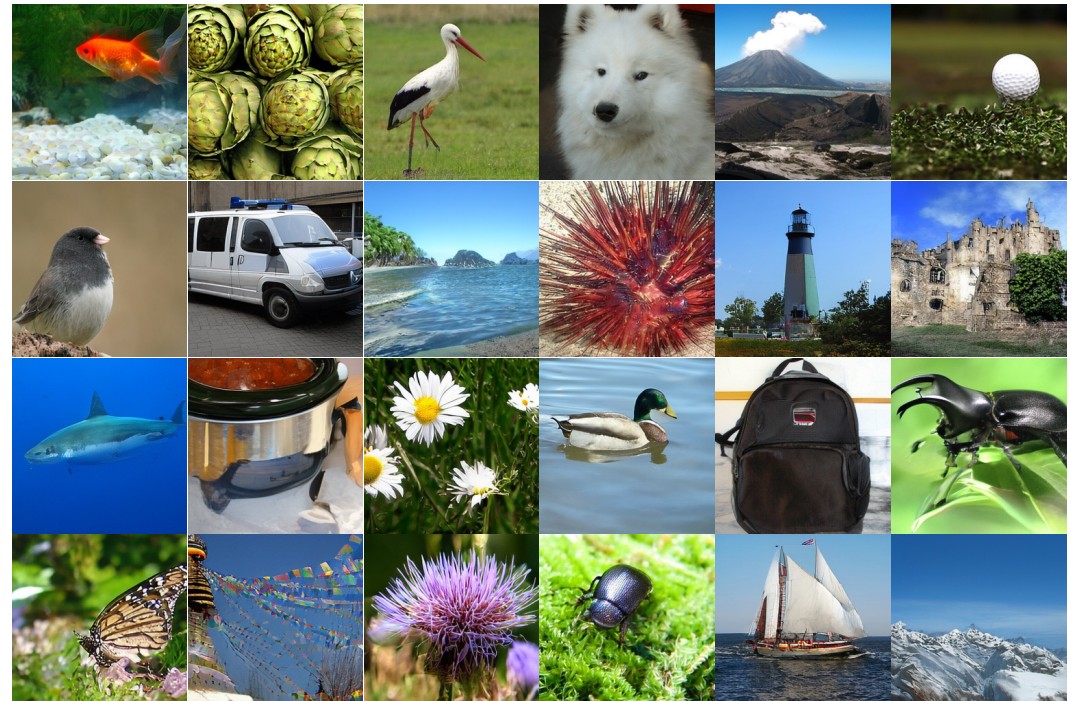

Figure 12: **Generated 256×256 samples by LaTtE-Flow Couple trained on ImageNet.**

## F  TIMESTEP-CONDITIONED RESIDUAL ATTENTION

Following the experimental setup in Section 6.2, we also perform an in-depth analysis on the LaTtE-Flow Blend variant. Figure 13 (a) shows how this sequential similarity across adjacent layers evolves over the sampling timesteps. The plot shows the mean similarity computed across 100 randomly sampled examples. We observe that for most of the adjacent layers, the sequential similarity is relatively low at early timesteps, and gradually increases as the timestep progresses, particularly in early layers, where the similarity rises and approaches 1.0. However, the observed similarity pattern varies significantly across timesteps and layers, motivating the need for a timestep-conditioned gating strategy of residual attention flows.

In Figure 13 (b), we visualize the learned residual attention gating values for head 11 within LaTtE-Flow Blend. These gates are dynamically modulated by timestep embeddings and control the degree to which residual attention from the previous layer is incorporated into the current layer's computation. To further understand the role of residual attention across heads, Figure 15 displays the gating values for all 12 heads in LaTtE-Flow Blend. We observe that gating remains relatively stable across timesteps within a specific head, but the patterns differ notably among different heads. A similar trend is also observed in the LaTtE-Flow variant (Figure 14), where head-specific gating patterns reflect different behaviors. In summary, these results validate the design of timestep-conditioned, head-specific residual attention. The gating mechanism enables adaptive reuse of earlier attention.

## G  THE USE OF LARGE LANGUAGE MODELS

In preparing this manuscript, we mainly used large language models (LLMs) as an auxiliary tool for polishing the writing. Specifically, the models were employed to improve sentence fluency, correct grammar errors, and refine clarity of expression. They were not involved in research ideation, experimental design, analysis, or substantive content generation.

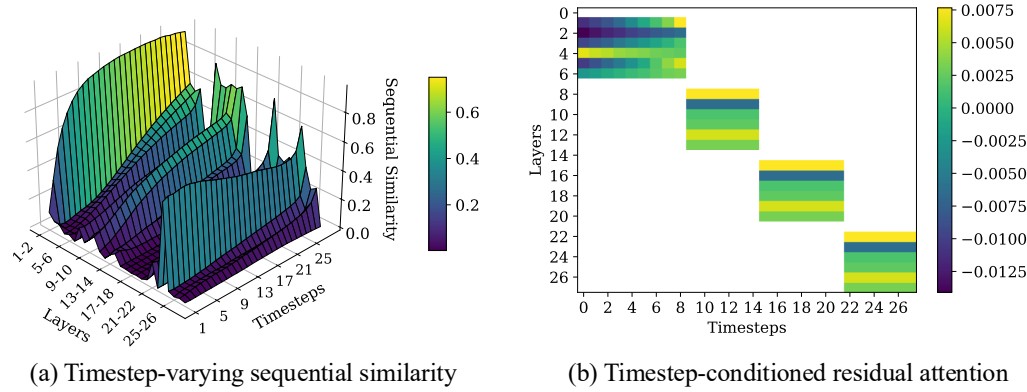

(a) Timestep-varying sequential similarity across adjacent transformer layers

(b) Timestep-conditioned residual attention gate for head 11 across transformer layers

Figure 13: **Visualization of attention in Baseline Blend and LaTtE-Flow Blend.** (a) Sequential similarity between adjacent layers increases over timesteps, particularly in early layers. (b) Residual attention gating in LaTtE-Flow Blend (head 11) shows relatively consistent gating values across timesteps within the same head.

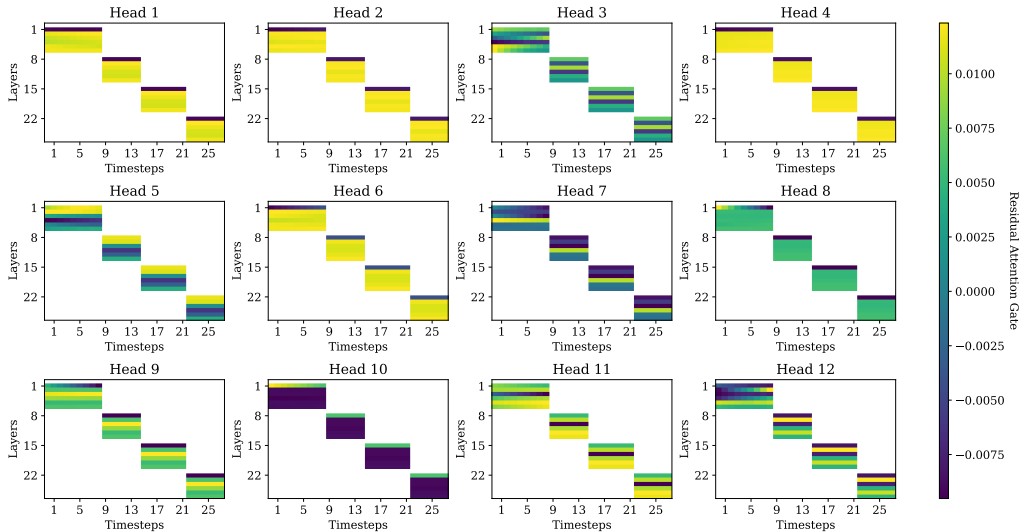

Figure 14: **Timestep-conditioned residual attention gates across transformer layer in LaTtE-Flow.** White regions indicate positions without gating values since residual attention is applied only within predefined layer groups. Notably, different heads exhibit distinct gating dynamics, with some emphasizing earlier timesteps, while others modulate more strongly in later layers, suggesting head-specific specialization in residual attention.

## H   IMPACT STATEMENT

This work advances the field of unified multimodal modeling by introducing LaTtE-Flow, an architecture that effectively combines image understanding and generation within a single, efficient framework. By leveraging pretrained vision-language models and introducing novel architectural mechanisms, Layerwise Timestep Experts and Timestep-Conditioned Residual Attention, LaTtE-Flow achieves strong performance with significantly improved inference speed. The proposed model has a potential impact in both academic and practical settings, as a scalable solution for building efficient, unified multimodal foundation models. It enables more efficient deployment of multimodal systems in resource-constrained environments, such as mobile devices or real-time applications, while maintaining high performance. While LaTtE-Flow improves performance and efficiency, it inherits the biases of its pretrained vision-language foundation and may generate misleading or in-

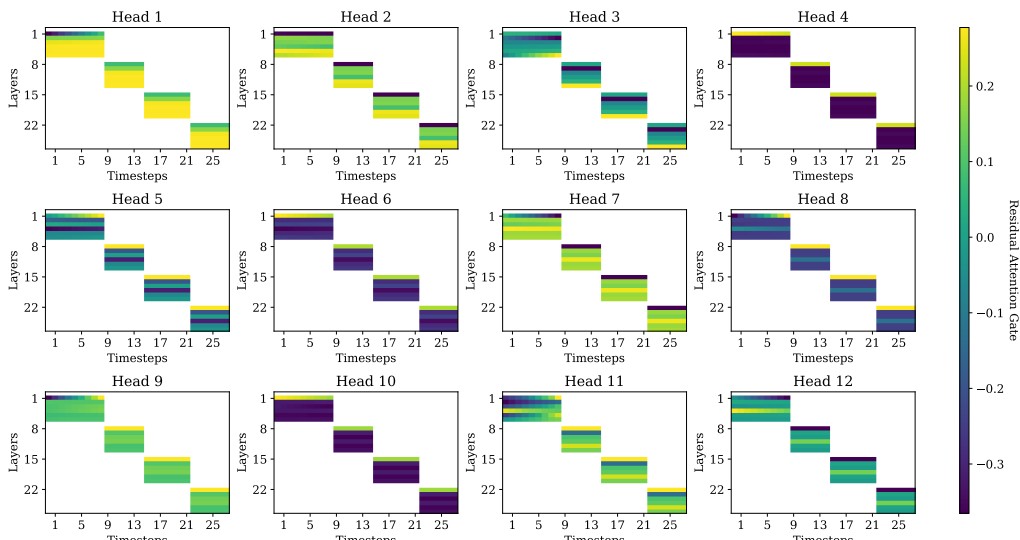

Figure 15: **Timestep-conditioned residual attention gates across transformer layer in LaTtE-Flow Blend.** White regions indicate positions without gating values since residual attention is applied only within predefined layer groups. Notably, different heads exhibit distinct gating dynamics, with some emphasizing earlier timesteps, while others modulate more strongly in later layers, suggesting head-specific specialization in residual attention.

appropriate outputs if not properly constrained. Careful evaluation and mitigation of such risks are important for downstream deployment.

## I  LIMITATIONS

Although LaTtE-Flow achieves substantial improvements in sampling efficiency with strong results in multimodal understanding and generation tasks, several limitations remain. First, our experiments involved training LaTtE-Flow for only 240K optimization steps, significantly fewer than existing unified multimodal models. Extending the training duration could potentially enhance the model's performance further. Second, while our uniform timestep distribution with overlapping intervals proved effective, the optimal timestep distributions or layer partitioning strategies remain an open problem. Future work should systematically explore and optimize timestep partitioning strategies.

