# OpenReview forum: "LaTtE-Flow: Layerwise Timestep-Expert Flow-based Transformer"
_ICLR.cc/2026/Conference — Submitted to ICLR 2026_

### Official Review · Reviewer_BahB · 2025-10-21

**Soundness:** 3
**Presentation:** 2
**Contribution:** 3
**Rating:** 6
**Confidence:** 3

**Summary:**

The paper proposes LaTtE-Flow, a novel architecture for extending image understanding models to perform image generation. Its core innovations are the Layerwise Timestep Experts (LTE), which partition transformer layers into groups specialized for different flow-matching timesteps to reduce inference computation, and Timestep-Conditioned Residual Attention (TCRA), a mechanism for reusing attention maps across layers. The method significantly reduces inference computational cost by using different activation layers on different time steps, while maintaining and even improving on both image generation and image understanding.

The proposed method is novel and seems to do lower computational cost at inference without sacrificing the performance. I suggest a weak accept.

**Strengths:**

1: The idea of LaTtE-flow is simple but effective. Based on its extension nature, it may be applied on any VLMs and trains them to also perform image generation with low computational cost.

2: LaTtE-flow yields better image generation and image understanding compared with advanced models of similar scale at much fewer activation parameters and average inference time.

3: The paper delivers a thorough ablation study.

**Weaknesses:**

1: The authors do not include the performance of the base model, Qwen2-VL-2B in Table 2, making it hard to compare how much gain LaTtE-Flow gives to the image understanding task.

2: The description of Table 2 does not match the content. It says "we report the number of activated parameters", however it doesn't. It also does not perform the computation cost(or time cost) for different models like in Table 1.

3: As an extension method, the authors only experiment LaTtE-Flow on one base model, Qwen2-VL-2B and does not generalise the method to other VLMs.

**Questions:**

1: Could you perform a more complete Table 2, with baseline performance(base model Qwen2-VL-2B) and computational cost( FLops, or other metrics)?
2: Is LaTtE-Flow able to perform on other base models besides Qwen2-VL-2B? How does it perform?

---

> ### Author Response · Authors · 2025-11-22
>
> **W1: Comparison with the base model / Q1: Table 2**
>
> Our proposed LaTtE-Flow uses Qwen2-VL-2B-Instruct as the understanding backbone, and we keep its parameters frozen to preserve its strong understanding capabilities (L311-L314). This design choice is consistent with recent works, including LMFusion [1], UniWorld [2] and BLIP3-o [3], which also utilize frozen backbones to maintain the pretrained understanding strength (L311-L314).
> Because the multimodal understanding component of our model is identical to Qwen2-VL-2B-Instruct, it achieves the same understanding performance as the backbone. For this reason, we omitted the Qwen2-VL-2B-Instruct results in Table 2.
>
> [1] LMFusion: Adapting Pretrained Language Models for Multimodal Generation
>
> [2] UniWorld-V1: High-Resolution Semantic Encoders for Unified Visual Understanding and Generation
>
> [3] BLIP3-o: A Family of Fully Open Unified Multimodal Models-Architecture, Training and Dataset
>
> **W2: Table 2 Content / Q1: Table 2**
>
> We thank the reviewer for pointing out this. To match the description and allow for a more complete comparison across models, we have updated Table 2 by adding two additional columns (highlighted in blue): (1) the number of activated parameters, and (2) the computation cost (measured in inference latency). Please refer to the revised version.
>
>
>
> **W3/Q1: Generalization to other VLMs**
>
> We thank the reviewer for the thoughtful questions. To further demonstrate that LaTtE-Flow is not tied to a specific VLM backbone, we conducted an additional experiment comparing two variants of LaTtE-Flow: (1) one with the generative pathway initialized from the pretrained Qwen2-VL model, and (2) another initialized from scratch (i.e., without any pretrained weights). We keep all hyperparameters identical. Due to time constraints during the rebuttal period, we trained the scratch-initialized version for 40K steps and evaluated performance on ImageNet 50K.
> We found that both models achieved comparable performance at the same training step, indicating that LaTtE-Flow is not constrained to any particular pretrained VLM for initialization.
>
> | **FID (↓)**     | **LaTtE-Flow** | **LaTtE-Flow (Scratch VLM)** |
> |------------------|------------------------|--------------------------------------|
> | 20K step         | 19.14                  | 21.67                                |
> | 40K step         | 12.23                  | 14.10                                |
>
>
> Finally,  to show that LaTtE-Flow is generally applicable to any Transformer-based diffusion or flow-based model, Appendix A provides an additional experiment on LaTtE-Flow Blend, an alternative unified architecture that merges understanding and generation computations within shared layers, similar to TransFusion. Our experiments demonstrate that both LaTtE-Flow variants outperform their respective vanilla baselines.

---

> ### Comment · Reviewer_BahB · 2025-11-23
> **Updated Table does not Include Base Model Performance**
>
> Please include the performance of base model (Qwen2-VL-2B) in performance comparison(table 1 and 2) so that we can see how much Latte-Flow gains.

---

> > ### Author Response · Authors · 2025-11-23
> >
> > Thank you for the quick response!
> >
> > We have included the performance of Qwen2-VL-2B-Instruct in Table 2. Since Qwen2-VL-2B-base is a pretraining model without instruction-following capability, we report results for its instruction-tuned counterpart, Qwen2-VL-2B-Instruct. In addition, because Qwen2-VL-2B-Instruct does not support image generation, it is not included in the comparisons in Table 1.
> >
> > We would also like to point out that our proposed LaTtE-Flow uses Qwen2-VL-2B-Instruct as the multimodal understanding backbone, and we keep its parameters frozen to preserve its strong understanding capabilities (L311-L314). This design choice is consistent with recent works, including LMFusion [1], UniWorld [2], and BLIP3-o [3], which also utilize frozen backbones to maintain the pretrained understanding strength. (L311-L314)
> > Because the multimodal understanding component of our model is identical to Qwen2-VL-2B-Instruct, LaTtE-Flow achieves the same understanding performance reported for Qwen2-VL-2B-Instruct in Table 2.
> >
> > Please let us know if this fully addresses your question or if any concerns remain. Thank you so much for your time and engagement in the discussion.
> >
> > [1] LMFusion: Adapting Pretrained Language Models for Multimodal Generation
> >
> > [2] UniWorld-V1: High-Resolution Semantic Encoders for Unified Visual Understanding and Generation
> >
> > [3] BLIP3-o: A Family of Fully Open Unified Multimodal Models-Architecture, Training and Dataset

---

> > > ### Comment · Reviewer_BahB · 2025-11-23
> > > **Thanks for the Response**
> > >
> > > Thanks for the clarification.
> > >
> > > So instead of improving on both image understanding and image generation, Latte-Flow only expands an image understanding model to the task of image generation.
> > >
> > > Based on this, I think Table 2 shall not be stated as a main contribution as Latte-Flow actually contributes nothing on this task.
> > >
> > > I notice that there is not a human preference evaluation on generative tasks of Table 1. Do you have human preference evaluation results for this task? Automated metrics like FID for image generative tasks are likely to overfit so this is necessary to prove real improvement.

---

> ### Author Response · Authors · 2025-11-24
>
> We thank the reviewer for the quick response! Our work follows recent unified model designs such as UniWorld-V1 [1] and BLIP3-o [2], which also employ frozen backbones to maintain the pretrained understanding capabilities of the VLM. In line with these works, our approach is designed to preserve the underlying VLM strengths. We have updated the caption in Table 2 to explicitly state that the model’s understanding ability is retained, and our user study is currently underway; we will share the results as soon as they are available. Thank you!
>
> [1] UniWorld-V1: High-Resolution Semantic Encoders for Unified Visual Understanding and Generation
>
> [2] BLIP3-o: A Family of Fully Open Unified Multimodal Models-Architecture, Training and Dataset

---

> ### Author Response · Authors · 2025-11-27
>
> Dear reviewer BahB,
>
> We have now conducted the user study comparing our model with other unified models.
> Specifically, we perform human preference comparisons among Janus Pro [1], Show-o [2], and our LaTtE-Flow.
> We randomly sample 50 prompts from ImageNet and collect image generations from the three models. For each prompt, the corresponding three images are displayed to 10 human annotators in randomized order, and annotators are asked to select the image they prefer.
> We instruct annotators to pay attention to (1) *Photo-realism*, and (2) *Semantic accuracy* (how faithfully the image depicts the target object). Detailed instructions are provided in Appendix D.
>
> The table below summarizes the win rates of our model against the baselines. LaTtE-Flow is preferred to Janus Pro in 71.4% of cases (with 8.6% ties and 19.6% losses), and preferred to Show-o in 63.4% of cases (with 5.0% ties and 31.4% losses). These results demonstrate that LaTtE-Flow consistently produces images that are more realistic and semantically accurate, indicating that the improvements brought by our unified modeling framework translate into meaningful gains in generative quality beyond automated metrics such as FID.
>
>
> | Comparison              | Wins (%) | Ties (%) | Losses (%) |
> |------------------------|----------|----------|------------|
> | **Ours vs. Janus Pro** | **71.4** | 8.6      | 19.6       |
> | **Ours vs. Show-o**    | **63.4** | 5.0      | 31.4       |
>
> Additional qualitative examples and details can now be found in Appendix D.
>
> Moreover, we have updated the caption of Table 2 to emphasize that LaTtE-Flow preserves strong image understanding, consistent with unified architectures.
>
> [1] Janus-Pro: Unified Multimodal Understanding and Generation with Data and Model Scaling
>
> [2] Show-o: One Single Transformer to Unify Multimodal Understanding and Generation
>
>
> We hope that our clarifications and additional results will be helpful as you form your final evaluation. We would be grateful if you could let us know if there are any remaining concerns, and we very much appreciate the time and effort you have invested in reviewing our work.

---

> > ### Comment · Reviewer_BahB · 2025-11-27
> > **Thanks for the Response**
> >
> > Thanks for the additional human preference experiment. I've decided to raise the score back to 6.

---

> > > ### Author Response · Authors · 2025-11-27
> > >
> > > Thank you very much for the update and for reconsidering our work! We truly appreciate your time, engagement, and constructive feedback throughout the discussion. Please let us know if there are any remaining questions or concerns, we would be more than happy to address them. We sincerely thank you again for considering our responses.

---

### Official Review · Reviewer_6gpc · 2025-10-28

**Soundness:** 3
**Presentation:** 3
**Contribution:** 3
**Rating:** 6
**Confidence:** 3

**Summary:**

This paper presents LaTtE-Flow, a unified MLLM which incorporates flow-matching-based image generation with VLM. First, they propose Layerwise Timestep Expert, distributing transformer layers into different timestep-specific experts, to improve the inference efficiency of unified MLLM. Second, they propose a gate attention approach to reuse the attention map in previous layers. Experiment results show that LaTtE-Flow outperform previous unified MLLM while being more efficient.

**Strengths:**

1. **The idea is interesting.** The idea of decouple multiple flow matching steps into multiple transformer blocks is interesting and results in good performance.

2. **Efficient.** LaTtE-Flow is very efficient, 6 times faster than Janus Pro. The author provides real running time to verify their claim.

3. **Effective.** LaTtE-Flow achieves low latency while keeping strong image understanding and generation performance.

**Weaknesses:**

1. **Compare and discuss with concurrent works.** Although using different data and model size, I suggest the author compare and discuss with newer Unified MLLM, including LMFusion, Blip3o and Bagel.

2. **Unification of generation and understanding.** LaTtE-Flow use different visual encoders and different sets of parameters for image understanding and generation. If the model first generates an image and then performs VQA based on the generated image, it requires two forward passes. I hope the authors can discuss this scenario.

3. **Lack of scale-up experiments.** This paper reports experiments only on 2B models. This is understandable, possibly due to computational constraints. It would be better if the authors could also run an 8B-scale model to demonstrate scalability.

**Questions:**

See weakness

---

> ### Author Response · Authors · 2025-11-22
>
> **W1: Comparison to concurrent works**
>
>
> We thank the reviewer for the helpful suggestion. Below, we provide a clearer discussion of how our work compares to LMFusion [1], Bagel [2], and Blip3o [3].
>
> LMFusion uses modality-specific FFNs and QKV projections to process text and image data separately. Both the text-specific and image-specific Transformer modules are initialized from the pretrained Llama-3-8B model, with the text module kept frozen during training. LMFusion is trained on approximately 0.25T image tokens. And the training data used is 380M proprietary Shutterstock image–caption pairs. Its goal is to efficiently reuse a pretrained LLM to improve multimodal understanding and generation.
>
> Bagel employs a Mixture-of-Transformer-Transformer architecture similar to LMFusion and is trained at a substantially larger scale. It is initialized from Qwen-2.5-1.5B/7B and is trained on ~5.1T multimodal tokens, including large interleaved image–text–video corpora, to study emergent capabilities at scale.
>
> BLIP3-o uses a decoupled auto-regressive + diffusion design. The Qwen-2.5-VL-7B-Instruct backbone is frozen as well to preserve the understanding performance, and the model takes in a learnable query that becomes the multimodal condition through the self-attention layers. The output multimodal condition is then input into the pre-trained Stable Diffusion XL model [4], which generates the output images. Blip3-o is trained on 25M open-source images plus 30M proprietary images, with additional instruction tuning.
>
> These works primarily target state-of-the-art multimodal understanding and generation quality through architectural scaling, large-scale data construction, and large-scale pre-training.
> In contrast, our work introduces two novel components that improve training and inference efficiency while offering competitive performance.
> LaTtE-Flow is orthogonal and complementary to the above approaches, and can in principle be incorporated into many transformer-based unified architectures that integrate flow-matching or diffusion models.
>
> We also provide a comparison table summarizing model size, training data, compute scale, inference cost (TFLOPs), and research focus.
>
>
>
> | Model        | Architecture                                              | Backbone Init.                               | Training Data Scale                                | Training Tokens | Inference Cost                                                                 | Primary Focus                                                    |
> |--------------|------------------------------------------------------------|------------------------------------------------|-----------------------------------------------------|------------------|----------------------------------------------------------------------------------|------------------------------------------------------------------|
> | LMFusion [1] | Modality-specific FFNs + QKV; shared attention; frozen text LLM | Llama-3-8B                                     | 380M proprietary image–caption pairs                | ~0.25T tokens    | 163.8 TFLOP                                       | Efficient adaptation of pretrained LLMs to multimodal unified models |
> | Bagel [2]    | Modality-specific Mixture-of-Transformer; shared attention | Qwen-2.5-1.5B / 7B                              | 26.665 B mixed text and image data                                                   | ~5.1T tokens     | 1.5B: 30.7 TFLOPs; 7B: 143.4 TFLOPs                | Scaling unified models for emergent multimodal performance       |
> | BLIP3-o [3]  | Frozen MLLM + diffusion decoder                           | Qwen-2.5-VL-3B-Instr. / 7B-Instr. + 3.5B Stable Diffusion XL | 25M open + 30M proprietary images + instruction tuning | ~0.11T tokens    | 3B: 14.0 TFLOPs; 7B: 14.6 TFLOPs | Improved multimodal understanding & image generation             |
> | LaTtE-Flow (ours) | Modality-specific modules; Layerwise Timestep-Expert Flow-based Transformer | Qwen2-VL-2B-Instr. | 1.2M ImageNet | ~0.03T tokens | 2.6 TFLOPs | Better sampling efficiency; competitive performance in unified models |
>
>
> [1] LMFusion: Adapting Pretrained Language Models for Multimodal Generation
>
> [2] Emerging Properties in Unified Multimodal Pretraining
>
> [3] BLIP3-o: A Family of Fully Open Unified Multimodal Models—Architecture, Training and Dataset
>
> [4] SDXL: Improving Latent Diffusion Models for High-Resolution Image Synthesis

---

> > ### Author Response · Authors · 2025-11-22
> >
> > **W2: Unification of generation and understanding**
> >
> > We thank the reviewer for raising this insightful point. Similar to LMFusion and Bagel, LaTtE-Flow employs different visual encoders and modality-specific parameters for image understanding and image generation. This design choice is motivated by empirical observation from previous papers [1,2,3] that using a single encoder—whether a VAE or a ViT—for both generation and understanding leads to noticeable performance degradation in one of the tasks. VAE works best for generation and ViT works better for understanding tasks.
> >
> > As a result, in scenarios where the model first generates an image and subsequently performs a VQA task on that generated image, all these systems must re-encode the generated image with the understanding encoder. This requirement is also explicitly shown in Figure 15 of Bagel.
> >
> >
> > We agree that this design introduces redundant computation, and achieving a truly unified visual representation capable of supporting both generation and understanding within a single pipeline is a compelling and important direction for future research. However, this exploration lies outside the scope of the current work.
> >
> > [1] LMFusion: Adapting Pretrained Language Models for Multimodal Generation
> >
> > [2] Emerging Properties in Unified Multimodal Pretraining
> >
> > [3] BLIP3-o: A Family of Fully Open Unified Multimodal Models—Architecture, Training and Dataset

---

> > > ### Author Response · Authors · 2025-11-22
> > >
> > > **W3: Scale-up Experiments**
> > >
> > >
> > > We thank the reviewer for the thoughtful comment regarding model scaling. To show the scalability of our method, during the rebuttal period, we trained a LaTtE-Flow variant built on Qwen2-VL-2B-Instruct as well as a larger LaTtE-Flow variant based on Qwen2-VL-7B-Instruct using the same training setup. Specifically, we train both model variants on Blip3o [1] data.
> > > Due to the limited time and computational resources available for rebuttal experiments, we have managed to train the 2B variant for 200K steps and the 7B variant for 100K steps.
> > > Below, we compare the performance of the 2B and 7B variants evaluated at the same training step. The 7B model exhibits consistent and meaningful improvements over the 2B variant on both COCO-30K FID and GenEval [2] on 100K steps, and outperforms the 2B variant evaluated on 200K steps, demonstrating clear and meaningful gains from increased model capacity.
> > >
> > >
> > > | Model             | Training Steps | COCO-30K FID (↓) | GenEval (↑) |
> > > |-------------------|----------------|------------------|-------------|
> > > | LaTtE-Flow (2B)   | 100K steps     | 35.91            | 31.5        |
> > > | LaTtE-Flow (2B)   | 200K steps     | **23.56**        | 40.0        |
> > > | LaTtE-Flow (7B)   | 100K steps     | 25.10            | **42.6**    |
> > >
> > >
> > > [1] BLIP3-o: A Family of Fully Open Unified Multimodal Models—Architecture, Training and Dataset
> > >
> > > [2] Geneval: An object-focused framework for evaluating text-to-image alignment.

---

> ### Comment · Reviewer_6gpc · 2025-11-25
> **Response to Author Rebuttal**
>
> Thanks for the author's response. The author addressed most of my concerns. After reading other reviewers' responses, I also have some concerns about the limited text-to-image generation experiments and performance, although it's reasonable due to limited computational resources.
> Overall, I lean to keep my score for this paper at this stage.

---

> > ### Author Response · Authors · 2025-11-27
> >
> > Dear Reviewer 6gpc,
> >
> > We sincerely thank the reviewer for the thoughtful follow-up and for acknowledging that most of the earlier concerns have been addressed. We also understand and appreciate the remaining concern regarding the paper scope and performance of our text-to-image experiments, especially in light of large-scale unified models.
> > Within our computational constraints, we have tried to provide as much evidence as possible that the proposed Layerwise Timestep-Expert and Timestep-Conditioned Residual Attention mechanisms yield real and robust improvements in efficiency and quality:
> > - In terms of efficiency, LaTtE-Flow achieves substantially faster inference, running 48× faster than Show-o and 6× faster than Janus Pro. Moreover, LaTtE-Flow also outperforms its corresponding Vanilla baselines, which are conceptually similar to LMFusion, while using significantly fewer activated parameters per flow-matching step and delivering 3× faster inference. The computational cost of the Vanilla baseline is 28.3 TFLOPs per forward pass, compared to only 7.08 TFLOPs for LaTtE-Flow, further underscoring the efficiency benefits of our proposed method.
> > - On ImageNet text-to-image generation, LaTtE-Flow improves both FID and convergence speed over the corresponding vanilla backbone under the same architecture and compute budget. Compared to state-of-the-art unified models that are pretrained on the mixture of ImageNet and other large-scale image-caption datasets, LaTtE-Flow achieves better FID scores while achieving much faster inference speed.
> > - In addition, during the discussion phase, we conducted a **human preference study** comparing LaTtE-Flow with Janus Pro and Show-o on 50 ImageNet prompts (10 annotators per prompt). Annotators prefer LaTtE-Flow in 71.4% of comparisons vs. Janus Pro and 63.4% vs. Show-o, showing that our architectural changes translate into **perceptually meaningful improvements** beyond automated metrics.
> >
> > While the additional general text-to-image experiments requested during rebuttal have not yet reached the state-of-the-art performance under tight time and compute constraints, they clearly demonstrate the strong generalization potential of our method beyond ImageNet.
> >
> > We hope this additional context helps clarify the intent and scope of our contribution, and we are sincerely grateful for the reviewer’s careful assessment and constructive feedback.

---

### Official Review · Reviewer_B17S · 2025-11-01

**Soundness:** 3
**Presentation:** 3
**Contribution:** 3
**Rating:** 6
**Confidence:** 4

**Summary:**

This paper proposes LaTtE-Flow, a unified and efficient framework for multimodal large language models that jointly handle visual understanding and image generation. The key idea is to enhance generation efficiency without sacrificing understanding capability. To achieve this, LaTtE-Flow introduces a Layerwise Timestep-Expert (LTE) design, where different Transformer layer groups specialize in specific timesteps of the flow-based generation process—thus reducing redundant computation during sampling. Additionally, a Timestep-Conditioned Residual Attention mechanism enables effective information reuse across layers and timesteps, improving coherence and stability in generation. With these designs, LaTtE-Flow substantially accelerates the flow-based generation process while maintaining high-quality visual outputs and strong understanding performance, outperforming prior unified models in both accuracy and efficiency.

**Strengths:**

1. The paper presents a clear and well-motivated problem statement, effectively highlighting the efficiency–quality trade-off in unified multimodal generation and offering a logically coherent solution through a flow-based Transformer design.

2. The proposed Layerwise Timestep-Expert mechanism is both elegant and practical, significantly improving inference efficiency by activating only relevant Transformer layers at each timestep.

3. The integration of Timestep-Conditioned Residual Attention is innovative, allowing effective feature reuse across layers and timesteps, which enhances both generation quality and training stability.

**Weaknesses:**

1. The paper does not provide a direct comparison between the proposed LaTtE-Flow and the original VLM backbone on multimodal understanding tasks, leaving unclear how much the unified training or flow-based adaptation affects understanding performance.

2. The work lacks quantitative results on standard text-to-image generation benchmarks, which limits the evaluation of LaTtE-Flow’s true generative capability and generalization to open-ended visual synthesis.

3. Although the architecture introduces several novel components, the ablation studies are relatively insufficient — many key design choices, such as the number of timestep experts or the specific contribution of residual attention, are not systematically analyzed.

**Questions:**

N/A

---

> ### Author Response · Authors · 2025-11-22
>
> **W1: Comparison with original VLM backbone**
>
> Our proposed LaTtE-Flow uses Qwen2-VL-2B-Instruct as the understanding backbone, and we keep its parameters frozen to preserve its strong understanding capabilities (L311-L314). This design choice is consistent with recent works, including LMFusion [1], UniWorld [2], and BLIP3-o [3], which also utilize frozen backbones to maintain the pretrained understanding strength. (L311-L314)
> Because the multimodal understanding component of our model is identical to Qwen2-VL-2B-Instruct, it achieves the same understanding performance as the backbone. For this reason, we omitted the Qwen2-VL-2B-Instruct results in Table 2.
>
> [1] LMFusion: Adapting Pretrained Language Models for Multimodal Generation
>
> [2] UniWorld-V1: High-Resolution Semantic Encoders for Unified Visual Understanding and Generation
>
> [3] BLIP3-o: A Family of Fully Open Unified Multimodal Models-Architecture, Training and Dataset

---

> ### Author Response · Authors · 2025-11-22
>
> **W2: Experiments on Text-to-Image Generation**
>
> We thank the reviewer for raising this point. To further evaluate LaTtE-Flow in broader text-to-image scenarios, we conducted additional experiments during the rebuttal period:
>
> ***a) Additional Text-to-Image Experiments***
> We trained LaTtE-Flow on the BLIP-3o [1] dataset, a standard text-to-image generation dataset, and conducted evaluation with COCO-30K FID and GenEval [2] benchmark.
>
> Due to time and resource constraints during the rebuttal period, this model has so far been trained for only 200K steps with a batch size of 128, corresponding to approximately 1.64B training tokens. For comparison, prior works typically train on substantially larger scales—for example, LMFusion [3] is trained on roughly 250B tokens. Nevertheless, the results already show clear improvements as training progresses, suggesting strong generalization potential beyond ImageNet.
>
> ***Table 1: LaTtE-Flow (2B) Performance on Text-to-Image Generation***
>
> | Training Steps | COCO-30K FID (↓) | GenEval (↑) |
> |----------------|------------------|--------------|
> | 100K           | 35.91            | 31.5         |
> | 200K           | 23.56            | 40.0         |
>
>
> ***b) Scaling to a Larger Model***
>
> Furthermore, we also trained a larger LaTtE-Flow variant built on Qwen2-VL-7B-Instruct using the same training setup. When evaluated on the same number of training steps, the 7B model variants outperform the 2B version, demonstrating that scaling up model capacity leads to improved text-to-image generation quality.
>
> ***Table 2: LaTtE-Flow (7B) Performance on Text-to-Image Generation***
>
> | Training Steps | COCO-30K FID (↓) | GenEval (↑) |
> |----------------|------------------|--------------|
> | 100K           | 25.10            | 42.6         |
> We will continue training these models and will update the results as they progress during the rebuttal period.
>
> [1] BLIP3-o: A Family of Fully Open Unified Multimodal Models-Architecture, Training and Dataset
>
> [2] Geneval: An object-focused framework for evaluating text-to-image alignment.
>
> [3] LMFusion: Adapting Pretrained Language Models for Multimodal Generation

---

> ### Author Response · Authors · 2025-11-22
>
> **W3: Ablation Studies**
>
> We thank the reviewer for the opportunity to clarify and further highlight our extensive ablation studies.
> In Section 6.2, we have provided five (5) ablation studies to support our claims and examine design choices.
>
> ***Number of Timestep Experts K (i.e., Number of Expert Groups K).***
>
> The experiments of “Impact of Varying Group Size” (L395-410) provided the analysis of the effect of having different numbers of timestep experts (i.e., number of expert groups K).
>
> Specifically, we evaluate group sizes M = 4, 7, 14, which correspond to K=7, 4, 2 expert groups (i.e., timestep experts), respectively, given that the total number of layers is 28. When the number of groups K is 1 (i.e., group size M equals the total number of layers), the model reverts to a vanilla architecture. We show the trade-off between inference time and model performance in Figure 5.
> We observe that larger group sizes (fewer expert groups/fewer timestep experts) lead to improved generation quality (lower FID) due to greater modeling capacity per timestep. However, this comes at the cost of slower inference, as more layers are activated at each step. To balance efficiency and performance, we adopt the number of experts K = 4 (i.e., group size M = 7) as our default configuration.
>
> ***Effect of Timestep-Conditioned Residual Attention.***
>
> In the experiments of “Effect of Timestep-Conditioned Residual Attention” (L411-418), we provided an ablation of the Timestep-Conditioned Residual Attention module, showing that removing this module leads to a clear drop in performance by 30% in FID, highlighting its critical role.
>
> Moreover, in the experiments of “Timestep Condition in Residual Attention” (L426-474), we analyze the timestep conditioning in the residual path, showing that timestep-conditioned attention gates enable dynamic, head-specific reuse of prior-layer signals.
>
>
> Section 6.2 also includes two additional key ablations:
> - an analysis of **training dynamics** (L380-394), showing that LaTtE-Flow converges faster than vanilla baselines, supporting our hypothesis that distributing timesteps across specialized transformer layers mitigates slow convergence in diffusion models;
> - an analysis of **sampling steps and guidance scales** (L420-424), showing performance improves with higher values up to CFG=5 and 40 steps, providing actionable guidance for sampling parameter selection.
>
> We also welcome any additional ablations the reviewer would find valuable.

---

### Official Review · Reviewer_wLVs · 2025-11-04

**Soundness:** 2
**Presentation:** 2
**Contribution:** 2
**Rating:** 2
**Confidence:** 5

**Summary:**

The core idea of this paper is to distribute the timestep modelling of flow matching models to different transformer layers, improving inference speed by activating only a small subset of layers at each sampling timestep. Moreover, a Timestep-Conditioned Residual
Attention mechanism is proposed to incorporate attention results across timesteps groups. The proposed method is instantiated with a VLM (i.e., Qwen2.5VL-2B), in the context of unified multimodal models. Following LMFusion, the LLM part is frozen to preserve the understanding ability of the original VLM. For image generation, the generation layers are trained on the ImageNet dataset.

**Strengths:**

1. The paper is well written and easy to follow. The sampling efficiency of visual generation is a significant research question.

2. The solution of distributing timestep modelling across transformer layers is intuitive. And the proposed time-conditioned residual attention effectively incorporates cross-layer information, boosting convergence and overall performance.

3. Comprehensive studies on the design choices, such as expert groups and the effects of residual attention, are conducted in the experiments.

**Weaknesses:**

1. *Unclear Motivation:* As stated in the abstract, the paper studies unified multimodal models that struggle to achieve the same level
of performance compared to specialist models. However, the paper only addresses the problem of sampling efficiency, which seems to have digressed from the core issue of unified models.

2. *Experiments Are Incomprehensive:* Although the paper is for unified multimodal models that include both text and image, the image generation of the model is only trained and evaluated on the class-conditial generation dataset---ImageNet.

**Questions:**

The idea of Layerwise Timestep-Expert seems a universal solution to all diffusion/flow-matching models, would it be applicalble to a wider range of DiTs for visual generation?

---

> ### Author Response · Authors · 2025-11-22
>
> **W1: Motivation**
>
> We thank the reviewer for the comments and the opportunity to clarify our motivation. Our work is concentrated on addressing the efficiency challenges in unified multimodal understanding-and-generation models. While unified multimodal systems offer broad capability, these models typically require extensive training, and many suffer from slow image generation speeds, limiting their practical deployment in real-time or resource-constrained settings.
>
> LaTtE-Flow directly addresses this core bottleneck through strategies that are generally applicable to any Transformer-based diffusion or flow-based model. To demonstrate the architectural generality beyond the main design, we also include in Appendix A experiments on LaTtE-Flow Blend, an alternative unified architecture that merges understanding and generation computations within shared layers, similar to TransFusion. And we show that both LaTtE-Flow variants outperform their respective vanilla baselines
>
>
> The main contributions of the paper (L089-096) include: (1) proposing LaTtE-Flow, an efficient and unified architecture, (2) introducing the Layerwise Timestep-Expert (LTE) design and (3) Timestep-Conditioned Residual Attention (TCRA), and (4) demonstrating strong understanding and generation performance with up to 6$\times$ faster inference.
>
> We have revised the abstract and introduction (highlighted in blue) to more clearly articulate this motivation, and we welcome any further suggestions for clarification.

---

> > ### Author Response · Authors · 2025-11-22
> >
> > **W2: Experiments on Text-to-Image Generation**
> >
> > We thank the reviewer for raising this point. We would like to clarify that, in our ImageNet experiments, instead of using the class IDs, we use the corresponding natural-language label captions for both training and evaluation. Thus, our ImageNet setting can be considered as a text-to-image generation task. We have revised the manuscript to explicitly state this detail (L309-311).
> >
> > To further answer the reviewer’s comment and evaluate LaTtE-Flow in broader text-to-image scenarios, we conducted additional experiments during the rebuttal period:
> >
> > ***a) Additional Text-to-Image Experiments***
> > We trained LaTtE-Flow on the BLIP-3o [1] dataset, a standard text-to-image generation dataset, and conducted evaluation with COCO-30K FID and GenEval [2] benchmark.
> >
> > Due to time and resource constraints during the rebuttal period, this model has so far been trained for only 200K steps with a batch size of 128, corresponding to approximately 1.64B training tokens. For comparison, prior works typically train on substantially larger scales—for example, LMFusion [3] is trained on roughly 250B tokens. Nevertheless, the results already show clear improvements as training progresses, suggesting strong generalization potential beyond ImageNet.
> >
> > ***Table 1: LaTtE-Flow (2B) Performance on Text-to-Image Generation***
> >
> > | Training Steps | COCO-30K FID (↓) | GenEval (↑) |
> > |----------------|------------------|--------------|
> > | 100K           | 35.91            | 31.5         |
> > | 200K           | 23.56            | 40.0         |
> >
> >
> >
> > ***b) Scaling to a Larger Model***
> >
> > Furthermore, we also trained a larger LaTtE-Flow variant built on Qwen2-VL-7B-Instruct using the same training setup. When evaluated on the same number of training steps, the 7B model variants outperform the 2B version, demonstrating that scaling up model capacity leads to improved text-to-image generation quality.
> >
> > ***Table 2: LaTtE-Flow (7B) Performance on Text-to-Image Generation***
> >
> > | Training Steps | COCO-30K FID (↓) | GenEval (↑) |
> > |----------------|------------------|--------------|
> > | 100K           | 25.10            | 42.6         |
> >
> > We will continue training these models and will update the results as they progress during the rebuttal period.
> > In conclusion, we would like to emphasize that in our setup, the ImageNet experiments are already conducted in a text-to-image generation setting, as we use natural-language label captions rather than class IDs. Beyond ImageNet, during the rebuttal period, we further conducted two additional experiments to show that LaTtE-Flow can (1) generalize well to more diverse text-to-image datasets, and (2) its performance improves further as model capacity increases.
> > Moreover, we would like to reiterate that the contributions of our paper center on improving efficiency (around 6$\times$ faster inference), introducing two novel architectural components (Layerwise-timestep experts and timestep-conditioned residual attention), and demonstrating that these designs improve efficiency without sacrificing model performance.  Our proposed Laywerwise-timestep expert and timestep-conditioned residual attention designs are applicable across different unified architectures. As illustrated in Appendix A experiments, LaTtE-Flow performs effectively when instantiated to an alternative unified architecture.
> >
> >
> > [1] BLIP3-o: A Family of Fully Open Unified Multimodal Models-Architecture, Training and Dataset
> >
> > [2] Geneval: An object-focused framework for evaluating text-to-image alignment.
> >
> > [3] LMFusion: Adapting Pretrained Language Models for Multimodal Generation

---

> > > ### Author Response · Authors · 2025-11-22
> > >
> > > **Q1: Applicability of LaTtE-Flow to DiT-based Models**
> > >
> > > We thank the reviewer for the insightful observation. Indeed, the proposed LaTtE-Flow framework is broadly applicable to Transformer-based diffusion and flow-matching models, including DiT-style architectures for visual generation.
> > >
> > > LaTtE-Flow introduces two novel and practically impactful methods (Layerwise Timestep Experts and Timestep-Conditioned Residual Attention) that improve sampling efficiency and training convergence. Importantly, both are designed to be modular and lightweight: the timestep-expert mechanism modifies only the training objective by allowing expert layers to specialize by timestep (Eq. 3) without any architectural changes, and the proposed timestep-conditioned residual attention mechanism introduces a timestep-gated residual connection between attention maps across consecutive layers, enabling efficient reuse of attention patterns during generation.
> > >
> > > This enables LaTtE-Flow to be broadly applicable to transformer-based diffusion and flow models.
> > >
> > > To demonstrate the generality of our method, Appendix A reports results for LaTtE-Flow Blend, an alternative unified architecture to which we successfully integrate our novel Layerwise Timestep Experts and Timestep-Conditioned Residual Attention modules. Across both architectures, LaTtE-Flow variants consistently improve over their corresponding vanilla counterparts.

---

> ### Comment · Reviewer_wLVs · 2025-11-24
> **Response to Author Rebuttal**
>
> Thanks for the authors' clarification in the responses. The reviewer acknowledges the technical innovation of the layer-wise timestep modeling approach that effectively addresses a common efficiency bottleneck in diffusion and flow models. However, this contribution is largely orthogonal to the core challenges of unified multimodal modeling. And positioning the work within the context of “unified models” may be somewhat misleading, as the proposed technique does not directly advance or analyze the unification aspect itself.
>
> This mis-positioning is coupled with an experimental setup whose rationale becomes difficult to justify. The LaTtE-Flow, as a unified multimodal model working on text-to-image generation, is mainly compared with class-conditioning generation models, under the ImageNet benchmark, showing a performance that largely lags behind classic DiT architectures. And the authors did not compare with flow-based vision transformers like SiT. On the other hand, the authors have provided additional results on text-to-image generation, showing results that are far below mainstream unified models (e.g., janus, harmon, metaquery).
>
> The reviewer understands that the cost of training text-to-image generation models from scratch is prohibitive. However, within the context of unified modeling, the current submission lacks sufficient experimental validation to substantiate its claims. In particular, the unified-model narrative would require comprehensive comparisons against strong unified baselines, which are currently missing. To make the empirical evaluation more convincing, the reviewer suggests reframing the scope toward class-conditional generation and conducting rigorous experiments based on established diffusion or flow transformer baselines such as DiT and SiT. This would allow the proposed layer-wise timestep experts to be evaluated in a controlled setting where its efficiency benefits can be clearly demonstrated and fairly compared, without relying on incomplete or mismatched multimodal baselines.
>
> Since neither extending the model to large-scale text-to-image training nor reframing the whole narrative of the paper is possible within minor or limited revisions, the reviewer will insist on a rejection for the current submission, and recommend a major revision for upcoming conferences.

---

> > ### Author Response · Authors · 2025-11-24
> >
> > We thank the reviewer for the detailed follow-up and for recognizing the technical novelty of our layer-wise timestep modeling. We understand that the remaining concerns relate primarily to the *framing* of the paper, rather than about the soundness of the proposed mechanisms themselves.
> >
> > As stated in the paper, the core contribution of LaTtE-Flow is an architectural efficiency improvement for diffusion/flow-based transformer, which we demonstrate in a unified modeling setting. We believe our work is an important and complementary direction, as improving the efficiency of the underlying generative backbone is itself a meaningful and practically relevant goal.
> >
> >  The core contributions of LaTtE-Flow are:
> > - Layerwise Timestep Experts for adaptively allocating capacity across timesteps;
> > - Timestep-Conditioned Residual Attention for timestep-aware attention modulation;
> > - A controlled empirical study showing that these mechanisms improve sampling efficiency while maintaining competitive quality under the same architecture backbones and compute budgets.
> >
> > These contributions are already directly validated through controlled comparisons where LaTtE-Flow and its Vanilla counterpart share the same backbone, training data, and compute budget.  Under this controlled setting, LaTtE-Flow consistently yields better quality–efficiency trade-offs.
> >
> > In addition, the generation module in our unified model is already a flow-matching transformer that is architecturally aligned with DiT-style backbones. Our comparison between LaTtE-Flow and the Vanilla variants, which share the same backbone, data, and compute budget, can be viewed as the “DiT-style controlled experiment” the reviewer is asking for. The observed gains in the quality–efficiency trade-off therefore directly reflect the benefits of the proposed architectural mechanisms, within a controlled diffusion/flow-transformer setting analogous to DiT/SiT.
> >
> > We agree that the unified model framing in the introduction can be clarified to ensure it does not overstate the scope of the work. We have further refined the wording accordingly so that the architectural contribution is presented as the primary focus. We appreciate the suggestion and believe this resolves the framing mismatch raised in the comment.

---

> ### Comment · Reviewer_wLVs · 2025-11-28
> **Response to Author Rebuttal**
>
> Thanks for the follow-up clarification. The reviewer acknowledges the technical innovation that improves sampling efficiency and empirical study has been conducted on the MLLM backbones. However, the reviewer still believes positioning the work under the topic of unified models requires more extensive experimental validations on main-stream multimodal generation tasks like text-to-image generation (e.g., GenEval, DPG and T2I-CompBench), and possibly image editing, which would better probe the unification aspect of multimodal models.
>
> The reviewer acknowledges the authors' efforts in addressing the review comments and is inclined to slightly raise the score to 4. However, since addressing the major concern would require more than the limited revisions possible in the rebuttal, the reviewer insists on rejecting the current submission.

---

> > ### Author Response · Authors · 2025-12-04
> >
> > We thank the reviewer for acknowledging the technical innovation of our work, as well as the inclination to raise the score.
> > Our primary contribution is architectural efficiency, not establishing a new state-of-the-art or better unification for unified models. The goal of this paper is to answer: "How do we solve the critical inference latency bottleneck inherent to unified models?" Our work provides a specific solution (LaTtE-Flow) that achieves up to 6$\times$ faster inference while preserving the underlying model's dual capabilities (understanding and generation). We believe this is a substantial and timely contribution.
> >
> > We have now trained a new LaTtE-Flow (7B) variant for **380K** steps during the rebuttal phrase.  At 380K steps, our model has processed approximately **3.11B training tokens**. This is orders of magnitude less than comparable unified works; for instance, LMFusion is trained on roughly 250B tokens. Despite this limited training budget, LaTtE-Flow achieves a GenEval score of 50.0. This is already **competitive with, and slightly surpasses, established baselines like SEED-X** (Ge et al., 2024) [1], which reports a GenEval score of 49.0.
> >
> >
> > ***Table 1: LaTtE-Flow (7B) Performance on Text-to-Image Generation***
> >
> > | Training Steps | GenEval (↑) |
> > |----------------|--------------|
> > | 100K        | 42.6         |
> > | 380K     | 50.0         |
> >
> > These results strongly validate that LaTtE-Flow generalizes effectively beyond ImageNet even with limited compute.
> >
> > [1] SEED-X: Multimodal Models with Unified Multi-granularity Comprehension and Generation

---

### Author Response · Authors · 2025-11-22
**General Response**

We sincerely thank all reviewers for their thoughtful and constructive feedback. We are encouraged to see that several key strengths of our work were consistently recognized across the reviews.
- **Novelty and well-adjusted technical contributions**: Reviewers `wLVs`,`B17S`, `6gpc` and `BahB` found the proposed Layerwise Timestep-Expert and Timestep-Conditioned Residual Attention design intuitive, novel, and well-justified.  Reviewer `BahB` also noted that LaTtE-Flow, as an extension-style design, is potentially applicable to any VLM backbone.
- **Strong empirical efficiency and performance**: Reviewers `B17S`, `6gpc`, and `BahB` highlighted our model’s strong empirical results in improving the inference efficiency. Reviewer `6gpc` highlighted that LaTtE-Flow achieves up to 6× speed-up over Janus-Pro. In particular, reviewer `B17S` praised LaTteFlow’s practicality while Reviewers `6gpc` and `BahB` further recognize its effectiveness in accelerating inference while maintaining strong understanding and generation performance.
- **Thorough and comprehensive ablation study**: Reviewers `wLVs` and `BahB` appreciated the comprehensive and thorough ablation studies on design choices, such as expert groups and effects of residual attention.
- **Clarity of motivation and paper presentation**: Reviewer` B17S` noted that the paper presents a clear and well-motivated problem statement. Reviewers `wLVs` also recognize that the paper is well written and easy to follow.

We have carefully considered all the constructive feedback and have addressed each reviewer's questions individually. We invite the reviewers to consider our detailed responses, and we would be happy to clarify any remaining questions.

Thank you once again for your time and insights!

Best wishes,

Authors

---

### Author Response · Authors · 2025-12-04

Dear AC, SAC, and PCs,

We would like to sincerely thank you for managing the review process and the reviewers for their thoughtful and detailed feedback.

The submission currently has **three positive scores (6, 6, 6)** from Reviewers `B17S`, `6gpc`, and `BahB`. The sole **negative score (2)** from Reviewer `wLVs` stems from a concern about framing not soundness, novelty, or empirical validity. Reviewer `wLVs` explicitly **acknowledges the technical innovation and correctness** of the proposed layerwise timestep modeling and agreed that it **effectively addresses a common efficiency bottleneck** in diffusion/flow models
We have substantially clarified the scope and framing to directly address this issue.

Below, we summarize the core strengths of our paper and how we have dedicated significant effort to address all comments.

**Core Contributions**

**1. Layerwise Timestep-Expert (LTE) Architecture for Efficient Flow-based Generation**

LaTtE-Flow introduces a novel Layerwise Timestep-Expert design that partitions Transformer layers into timestep-specialized groups, so that only a subset of layers is active at each sampling step. This achieves an asymptotic **K-fold reduction in per-step compute**, where K denotes # timestep-expert groups, **while preserving generation quality**.

**2.    Timestep-Conditioned Residual Attention (TCRA) for Information Reuse**

We propose Timestep-Conditioned Residual Attention, which reuses self-attention maps from previous layers via a timestep-dependent gating mechanism, encouraging efficient refinement across layers. Ablations show that **removing TCRA significantly worsens both FID↓  and IS↑** by 42.6% (5.79 → 8.26) and by 26.3% (213.1 → 157.0), respectively. Precision and recall also drop, demonstrating that TCRA improves both sample fidelity and generative coverage.

**3. Substantial Efficiency Gains with Competitive Quality**

On ImageNet-50K, LaTtE-Flow attains competitive FID and IS while being **6× faster than Janus Pro** and **48× faster than Show-o**, and using significantly fewer activated parameters per step. Compared to its Vanilla backbone, LaTtE-Flow achieves **better quality–efficiency trade-offs and 3× speedup** under identical architecture and data. Our human evaluation study shows these improvements also translate into perceptually meaningful gains.

**4. Generality and Compatibility with Diffusion/Flow Transformers**

Our extensive experiments show applicability to both the mixture-of-transformers unified architecture in the main paper and an alternative “LaTtE-Flow Blend” variant (in Appendix) that merges understanding and generation computations within shared layers. Overall, experiments demonstrate that our new LTE/TRCA modules are not tied to a specific backbone design.

**Strengths Recognized by Reviewers**

Across reviews, there is strong agreement on several key strengths:

- **Novelty & Technical Soundness**. `wLVs` acknowledges the **technical innovation** of layerwise timestep modeling and that it **effectively addresses a common efficiency bottleneck in diffusion/flow models**. `B17S` finds the LTE mechanism **elegant and practical**, and TCRA **innovative** for feature reuse and stability. `6gpc` describes the idea as **interesting and effective**, noting the clear efficiency gains. `BahB` calls LaTtE-Flow **simple but effective**, with **potential to be applied to many VLMs** as an extension-style design.

- **Empirical Efficiency & Practicality**. `6gpc` highlights that LaTtE-Flow is **very efficient**, with 6× speedup over Janus Pro and real running-time measurements.
- **Clarity & Thorough Ablations**.`B17S and `wLVs` note that the paper is well-written and easy to follow. `BahB` and `wLVs` point to the **comprehensive ablation studies** on group size, residual attention, training dynamics, sampling steps, and CFG as a strength.

---

> ### Author Response · Authors · 2025-12-04
>
> Our rebuttal provided a substantial set of additional experiments and clarifications to systematically address every question and concern raised by the reviewers:
>
> 1. **Expanded Text-to-Image Evaluation**. New experiments of general text-to-image generation on GenEval demonstrate that LaTtE-Flow generalizes beyond ImageNet. These results directly address concerns about limited T2I evaluation.
>
> 2. **Scalability Demonstration (2B → 7B)**. A 7B LaTtE-Flow variant, trained under the same setup, outperforms the 2B model at comparable training steps, demonstrating **scaling behavior** and that the proposed mechanisms remain effective at larger scales.
>
> 3. **Human Preference Study**. A human evaluation compared LaTtE-Flow to Janus Pro and Show-o. **Annotators preferred LaTtE-Flow 71.4% vs Janus Pro and 63.4% vs Show-o**, focusing on photorealism and semantic accuracy. This confirms that the architectural improvements translate into **perceptually meaningful gains**.
>
> 4. **Clarified Framing and Scope**. We considerably revised the framing in the abstract and introduction to more clearly foreground the **core contribution of architectural efficiency improvements with demonstrated empirical benefits**. This directly responds to `wLVs`’ concern about *“mis-positioning.”* Furthermore, as detailed in our previous comments, we conducted additional general text-to-image generation experiments to directly respond to the reviewer's concerns regarding performance on general text-to-image generation.
>
> 5. **Controlled Experiments**. Our rebuttal clarified that the Vanilla backbone shares architecture, data, and training budget with LaTtE-Flow, providing a controlled experiment. The observed gains in FID and efficiency directly measure the benefit of LTE and TCRA.
>
> Given the strong positive majority, the clear technical novelty and soundness, and the extensive additional experiments that comprehensively address reviewer concerns, we believe that our contributions are timely and relevant for the ICLR community, offering broad methodological advances and practical insights for efficient, scalable multimodal generation and understanding.
>
> Thank you very much for your time and careful consideration in handling our submission.
>
> Best,
>
> The authors of LaTtE-Flow

---

### Meta-Review · Area_Chair_ofRZ · 2026-01-06

**Summary:**

This paper proposes the Layerwise Timestep Expert Flow-based Transformer (LaTtE-Flow) to improve the efficiency of the diffusion/flow-based transformers for unified models. The paper received scores of 6, 6, 6, and 2. The first concern of the reviewer wLVs regarding unclear motivation was addressed in the rebuttal. However, the second concern regarding the comprehensiveness of the experiments, which has also been raised by reviewer B17S and 6gpc, has not been well addressed.

After carefully reviewing the paper and rebuttal discussion, the AC agrees with the motivation and technical novelty of this paper. However, the AC also concurs with Reviewer wLVs that the current submission lacks sufficient experimental validation to support the claims. Thus, the AC recommends rejecting this paper and encourages the authors to add more experimental results and resubmit.

**Reviewer Concerns:**

The minor concerns (e.g., motivation, ablation studies, and comparisons with concurrent works etc.) were addressed during the rebuttal period. However, the major concern regarding the comprehensiveness of the experiments, which was raised by Reviewers wLVs, B17S, and 6gpc, remains outstanding.

**Reviewer Scores:**

While Reviewer wLVs raised the score from 2 to 4 after the concerns regarding motivation were addressed, the major concern regarding the comprehensiveness of the experiments remains outstanding. Consequently, it is unlikely that the other reviewers will further adjust their scores.

---

### Decision · Program_Chairs · 2026-01-26

Reject